# NF-κB subunits direct kinetically distinct transcriptional cascades in antigen receptor-activated B cells

Mingming Zhao [1,2], Prashant Chauhan [1], Cheryl A. Sherman [1], Amit Singh [1], Mary Kaileh [1], Krystyna Mazan-Mamczarz[3], Hongkai Ji [4], Jaimy Joy [1], Satabdi Nandi [1], Supriyo De [3], Yongqing Zhang[3], Jinshui Fan[3], Kevin G. Becker[3], Png Loke [2], Weiqiang Zhou [4] & Ranjan Sen [1]✉

The nuclear factor kappa B (NF-κB) family of transcription factors orchestrates signal-induced gene expression in diverse cell types. Cellular responses to NF-κB activation are regulated at the level of cell and signal specificity, as well as differential use of family members (subunit specificity). Here we used time-dependent multi-omics to investigate the selective functions of Rel and RelA, two closely related NF-κB proteins, in primary B lymphocytes activated via the B cell receptor. Despite large numbers of shared binding sites genome wide, Rel and RelA directed kinetically distinct cascades of gene expression in activated B cells. Single-cell RNA sequencing revealed marked heterogeneity of Rel- and RelA-specific responses, and sequential binding of these factors was not a major mechanism of protracted transcription. Moreover, nuclear co-expression of Rel and RelA led to functional antagonism between the factors. By rigorously identifying the target genes of each NF-κB subunit, these studies provide insights into exclusive functions of Rel and RelA in immunity and cancer.

The nuclear factor kappa B (NF-κB) family of transcription factors mediate responses to cell stimulation[1]. NF-κB responses have been most extensively studied in response to proinflammatory signals such as tumor necrosis factor alpha (TNFα), interleukin 1 beta (IL1β) and bacterial lipopolysaccharides (LPS)[2,3]. These agents induce NF-κB by the post-translational (classical) pathway via degradation of inhibitor of NF-κB (IκB) proteins. RelA, Rel and Nfkb1 family members are major mediators of classical NF-κB activation. Other stimuli, such as lymphotoxin β, B cell activating factor (BAFF) and CD40 activate the nonclassical pathway by elevating levels of NF-κB-inducing kinase (NIK). Thereafter, NIK-mediated phosphorylation and processing of Nfkb2 permit nuclear translocation of RelB-containing heterodimers for gene expression. Given the multitude of signals that activate it, a fundamental question in NF-κB biology is the basis by which the specificity of cellular responses is assured.

The specificity question can be parsed in three ways. First, how is cellular specificity achieved? This pertains to circumstances where the same signal induces NF-κB in different cell types, evoking different responses. Second, how is stimulus specificity achieved? This applies when distinct NF-κB stimuli induce NF-κB in the same cell type yet evoke distinct responses. Third, how is subunit specificity achieved? This is most pertinent in the comparison of Rel and RelA, both of which are activated by the classical pathway and coexpressed in most hematopoietic cells. The importance of subunit specificity is evident from the distinct phenotypes of *Rel*- or *Rela*-gene knockouts in mice[4–8].

NF-κB-dependent gene expression has been thoroughly explored in macrophages. Using stringent criteria, a study discussed in ref. 9 identified a subset of primary response genes that were dependent on RelA for expression in response to lipid A. A smaller subset of these

[1]Gene Regulation Section, Laboratory of Molecular Biology and Immunology, National Institute on Aging, Baltimore, MD, USA. [2]Type 2 Immunity Section, Laboratory of Parasitic Diseases National Institute of Allergy and Infectious Diseases, Bethesda, MD, USA. [3]Computational Biology and Genomics Core, Laboratory of Genetics and Genomics, National Institute on Aging, Baltimore, MD, USA. [4]Department of Biostatistics, Johns Hopkins University Bloomberg School of Public Health, Baltimore, MD, USA. ✉e-mail: senranja@grc.nia.nih.gov

bound RelA at their promoters and were proposed to be direct targets of RelA. Sites of RelA recruitment in LPS-activated macrophages also bind PU.1, a B- and macrophage-specific transcription factor, and are marked by DNase 1 hypersensitive sites before LPS treatment[10]. These observations have led to the working model that cell-specific responses are directed by cell-specific accessibility of NF-κB to selected sites in the genome. However, this reasonable model has not been systematically tested by comparing gene expression and genome-wide binding in different cell types. The basis for stimulus specificity of NF-κB-dependent responses has also been explored in the context of inflammatory signaling in macrophages. Accumulating evidence indicates that differences in kinetic patterns of NF-κB induction by specific stimuli contribute to the pattern of inducible gene expression[11,12].

By contrast, the basis for subunit specificity is least explored. According to a study discussed in ref. 13, careful analysis of the affinity of different NF-κB family members to related κB motifs suggests that subunit specificity could, in part, be achieved via distinct κB motifs in RelA- or Rel-specific target genes. Due to inadequate definition of RelA or Rel target genes, this model has also not been experimentally validated. It even remains unclear whether RelA and Rel bind to disparate κB sites in cells, as suggested by in vitro studies.

RelA and Rel have distinct functions in B cell biology, although neither protein is essential for B cell development. Rel deficiency alone, or in combination with Nfkb1, severely reduces germinal center (GC) formation during immune responses[14]. Thus, RelA cannot compensate for Rel in GC responses. Additionally, ectopic expression of transgenic Rel has been shown to increase GC responses[15]. NF-κB proteins, Rel in particular, have also been implicated in B cell lymphomas[16–18]. Despite involvement in critical physiological phenomena, NF-κB subunit specificity that distinguishes essential roles of Rel in GC responses or cancer is not known.

In this study, we combined time-dependent multi-omics to gain insights into the basis for subunit specificity of NF-κB family members in splenic B lymphocytes activated via the B cell receptor (BCR). RelA and Rel bound at early and late activation time points, respectively, and subsets of genomic-binding sites were selective for each factor. Bulk and single-cell RNA-sequencing (scRNA-seq) studies revealed marked heterogeneity of RelA- or Rel-specific gene expression and temporally distinct sets of target genes for each factor. The list of target genes identified included many that had not been previously linked to NF-κB. Additionally, we found that marginal zone (MZ) B cells were refractory to early NF-κB signaling, and nuclear co-expression of RelA and Rel resulted in mutual antagonism between the two factors. Our studies define kinetically distinct cascades of gene expression regulated by specific NF-κB subunits in BCR-activated B cells and thereby provide insights into exclusive functions of RelA and Rel in immunity and cancer.

## Results

NF-κB target genes are poorly defined, especially in primary untransformed cells. Additionally, the extent to which RelA or Rel compensate for each other to regulate gene expression has not been previously

characterized. To address these questions, we examined the responses of mouse splenic B cells activated via the BCR. We considered genes to be direct targets of NF-κB if the following criteria were observed: (1) they bound NF-κB in response to stimulation, (2) their expression changed in response to cell stimulation and (3) their inducible activity was altered by the absence of NF-κB (ref. 19). We applied these criteria to each NF-κB subunit to identify RelA- or Rel-specific target genes. A substantial subset of genes that satisfied only the latter two criteria was considered 'indirect' targets of NF-κB.

### Genome-wide recruitment of RelA and Rel

BCR crosslinking initiates two phases of NF-κB activation in mouse B splenic B cells[20]. The first transient phase occurs by translocation of both RelA and Rel from cytoplasmic pools. Nuclear levels of RelA peak around 1 h and are restored back to the cytoplasm by 4 h. The second phase is dominated by newly translated Rel, with nuclear levels increasing between 6 h and 24 h. To account for the biphasic response, we examined NF-κB binding, chromatin accessibility, histone modifications and gene expression at 1, 4 and 18 h after continuous BCR crosslinking.

We carried out chromatin immunoprecipitation followed by sequencing (ChIP–seq) with anti-RelA and anti-Rel antibodies in naïve B cells treated with anti-IgM (Fab'2). At optimal anti-IgM crosslinking conditions, we determined that approximately half the cells induced RelA nuclear translocation (Supplementary Fig. 1a,b). Specificity of each antibody for ChIP was substantiated by quantitative PCR at previously characterized NF-κB target genes (Supplementary Fig. 1c). After peak calling using CisGenome (Supplementary Fig. 1d), we focused on peaks that were present in two biological replicate experiments (Supplementary Fig. 1e). RelA was recruited to 3,821 sites, which were mostly located in intronic and intergenic regions, after 1 h BCR crosslinking of wild-type (WT; C57BL/6) splenic B cells (Fig. 1a, left). RelA-bound sites dropped sharply at 4 h, closely following the pattern of bulk nuclear RelA. Hypergeometric Optimization of Motif EnRichment[10] (HOMER, findMotifsGenome.pl program) analysis (Fig. 1a, bottom) identified motifs for Sfpi1, NF-κB-RelA and PU.1/IRF composite motifs as the top three sequences associated with regions of inducible RelA binding (see also Supplementary Fig. 1f). By contrast, maximum Rel binding (3,940 sites) occurred after 18 h of activation (Fig. 1a, right), also to intronic and intergenic regions that were enriched for NF-κB, Sfpi1 and RUNX binding motifs (Fig. 1a, bottom, and see also Supplementary Fig. 1g). HOMER analyses were substantiated using TFmotifview[21] (Supplementary Fig. 1h). Inducible Rel-binding sites at 1 h largely overlapped with RelA binding at 1 h (Supplementary Fig. 1i). We conclude that the first and second phases of NF-κB-induced signaling lead to genome-wide recruitment of RelA and Rel, respectively. More restricted recruitment of Rel at early time points occurred to sites that also bound RelA.

Approximately half (42%) of Rel-binding sites at 18 h overlapped RelA binding at 1 h (Fig. 1b); several thousand other sites were selective for either RelA or Rel. These assignments were validated for a subset of genes via independent ChIP followed by quantitative PCR assays (Fig. 1c). Genomic locations of RelA- and Rel-specific binding

---

**Fig. 1 | Identification of RelA- and Rel-binding sites in BCR-activated B cells.** **a**–**d**, Splenic B cells from C57BL/6 mice were activated with anti-IgM (F(ab)'2) for 0, 1, 4 and 18 h. ChIP was carried out using anti-RelA and anti-Rel antibodies. Coprecipitated genomic DNA was used to generate libraries for sequencing. Data from two independent ChIP–seq experiments were processed as described in Methods. Results shown are for RelA- and Rel-binding sites that were common to both replicates with threshold peak scores as described in Methods. **a**, Genomic locations of RelA (left) and Rel (right) binding sites across multiple genomic regions, as annotated by HOMER. Total number of peaks at each activation time point are noted above the bars. Top substantial transcription factor binding motifs enriched within RelA peaks (top panels) and Rel peaks (lower panels) at different time points were identified using HOMER (see also Supplementary

Fig. 1f,g). **b**, RelA and Rel ChIP–seq libraries were merged to identify shared and unique binding sites for each NF-κB subunit at each different time points (Venn diagram). Sequence motifs within RelA-specific, Rel-specific and RelA/Rel shared peaks are shown alongside (see also Supplementary Fig. 1k; **a** and **b**, Default statistical setting used to obtain *P* value by HOMER). **c**, Rel or RelA binding to select target genes was verified by ChIP–PCR. Data shown are the average of two additional ChIP experiments with cells activated for the indicated times; fold change = $2^{(CT(input)−CT(target))}/2^{(CT(input)−CT(neg))}$. **d**, Representative browser tracks based on mm9 annotation of ChIP–seq profiles of genes that bind only RelA (*Mcl1*), only Rel (*Psma6*) or both Rel and RelA (*Dennd4a*). The *y* axis represents normalized reads per 10 million aligned reads (see also Supplementary Fig. 1l).

sites were similar (Supplementary Fig. 1j). However, Rel-bound sites were enriched for motifs of IRF proteins (Supplementary Fig. 1k). Representative genome browser tracks are shown in Fig. 1d and

Supplementary Fig. 1l. We conclude that RelA or Rel recognizes similar sequences in vivo and many sites sequentially bind RelA (early) and Rel (late).

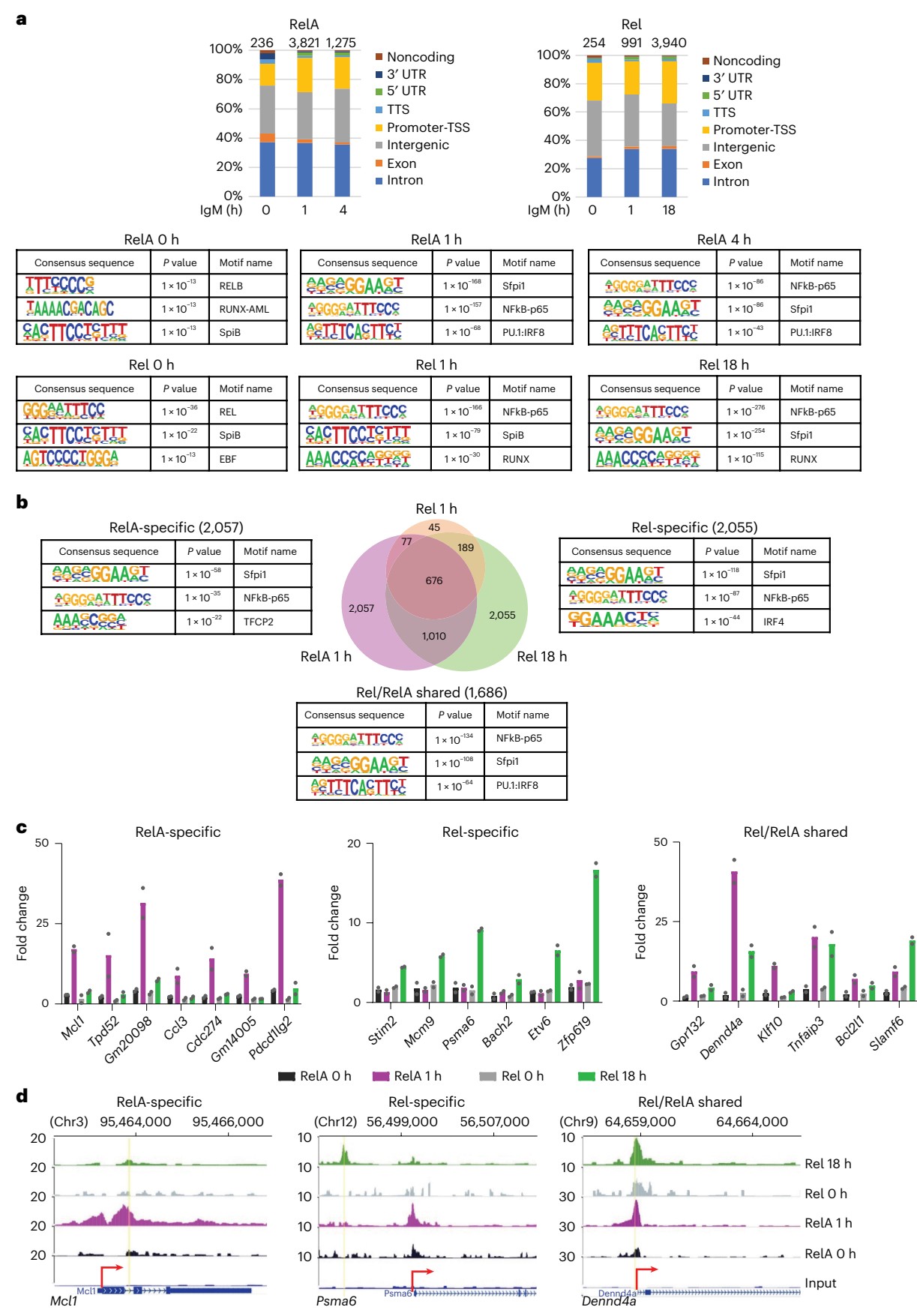

To examine epigenetic features of NF-κB responses, we carried out assay for transposase-accessible chromatin with sequencing (ATAC-seq) and histone ChIP−seq over the same time course (Supplementary Fig. 2a,b). Sites of both early RelA and late Rel recruitment were marked by accessible chromatin and histone modifications associated with active promoters (H3K4me3) or enhancers (H3K27ac) before cell activation (Fig. 2a,b). From time-dependent ATAC-seq, we identified several thousand sites at which pre-existing chromatin accessibility was transiently increased (Fig. 2c, top). Most of these sites bound RelA at 1 h and were not marked by H3K4me3, indicating that they fell in introns and intergenic regions. Transient ATAC sensitivity at these sites was consistent with changes being induced by transiently activated transcription factors such as RelA. An example is the *Irf4* gene (Fig. 2d, left). Increased chromatin accessibility with 18 h activation correlated with Rel binding to a subset of H3K4me3+ promoter sites (Fig. 2c,d, bottom and right, respectively). Inducible and constitutive ATAC sites were enriched for distinct transcription factor motifs (Supplementary Fig. 2c,d). We conclude that RelA or Rel binding to enhancers and promoters, respectively, can induce chromatin accessibility changes in response to BCR activation.

## Transcriptional responses to BCR stimulation

To decipher relationships between chromatin changes, RelA/Rel binding and gene expression, we carried out RNA-seq analyses of splenic B cells activated with anti-IgM (Supplementary Fig. 3a). We identified approximately 3,000 genes by EBSeq[22] that were either upregulated or downregulated more than twofold over the time course (FDR ≤ 0.05; Fig. 3a). The timing of gene expression changes mostly coincided with or followed changes in chromatin accessibility (Fig. 3b). Upregulated genes (≥2-fold change) were further classified by *k*-means clustering into three categories based on the peak expression at 1, 4 or 18 h (Fig. 3c). After annotating NF-κB binding to genes using HOMER (annotatePeak.pl algorithm), we observed inducible RelA binding to 39% of genes in category 'a' (rapidly and transiently induced) and 13% of genes in categories 'b' and 'c' (Extended Data Table 1). RelA binding decreased after 4 h of anti-IgM treatment, whereas numbers of Rel-bound genes increased at 18 h, concordant with the overall patterns of RelA and Rel ChIP−seq (Extended Data Table 1). For parity, downregulated genes (≤2-fold change) were also distributed into three categories that broadly coincided with rapid, intermediate and transient downregulation (Fig. 3c, labeled d, e and f). Both RelA and Rel bound to several hundred genes that were downregulated by anti-IgM treatment (Extended Data Table 1).

To identify NF-κB target genes, we carried out RNA-seq analyses with WT splenic B cells treated with anti-IgM in the presence of BAY 11-7082 (abbreviated as BAY hereafter), an inhibitor of IKK2 (ref. 23). By blocking classical NF-κB activation, the inhibitor suppressed BCR-induced RelA and Rel at early time points (Supplementary Fig. 3b, top). We found that BAY treatment also abrogated late Rel activation (Supplementary Fig. 3b, bottom), thereby permitting its use throughout the experimental time course. Differential gene expression analysis by DESeq2 followed by *k*-means clustering identified six kinetic patterns of gene expression (Fig. 3d, clusters I1−I6). A total of

186 transiently induced RNAs were suppressed by the inhibitor (Fig. 3d, cluster I3); 113 of these genes bound RelA at 1 h (red numbers), implicating them as direct NF-κB targets (Extended Data Table 2). A total of 101 of these genes also bound Rel at 18 h (blue numbers), although RNA levels did not change. A total of 271 genes with intermediate activation profiles were suppressed by the inhibitor (Fig. 3d, cluster I2), of which 64 bound RelA at 1 h. A total of 1,179 late-induced genes were suppressed by the inhibitor (Fig. 3d, clusters I1 and I6), of which 254 bound Rel at 18 h. Although many of these genes also bound RelA at 1 h, maximal RNA expression coincided with late Rel binding. For a small subset of target genes, we confirmed that RNA induction was accompanied by increased protein expression (Supplementary Fig. 3c). Taken together, these experiments identify early and late NF-κB target genes in BCR-activated B cells (Fig. 3g and Extended Data Tables 3−5).

In each cluster of differentially expressed genes, we found many genes that did not inducibly bind either RelA or Rel (Fig. 3d, difference between gray and red/blue numbers). Our interpretation is that such genes were activated by transcription factors induced by NF-κB and reflect transcriptional cascades initiated by NF-κB activation (see subsection 'NF-κB-induced cascades'). Kyoto Encyclopedia of Genes and Genomes (KEGG) pathway analyses revealed distinct biological functions of early and late NF-κB activation (Supplementary Fig. 3d). The early transient response (cluster I3) captured pathways such as 'NF-κB signaling' and various cancer-related pathways as the most significant. By contrast, long-term responses (clusters I1 and I6) were enriched for aspects of cell cycle regulation, metabolism and neurodegenerative diseases (Supplementary Fig. 3d). Finally, RelA and Rel also bound to many genes that were activated by BCR crosslinking but unaffected by the inhibitor (Supplementary Fig. 3e) or unaffected by activation (Supplementary Fig. 3f). Such genes may be activated by NF-κB in response to other stimuli.

## Subunit selective NF-κB target genes

To distinguish between the effects of RelA and Rel, we carried out time-dependent RNA-seq in RelA-(*RelA^{fl/fl}xCd19-cre* mice; Supplementary Fig. 4a) or Rel-(*Rel^{−/−}*) deficient B cells following BCR crosslinking. Previous studies have shown that germline or conditional Rel knockouts have similar B cell subsets and functional responses[24]. Differential gene expression analyses were carried out with DESeq2 and displayed after *k*-means clustering (Fig. 3e,f). The most prominent effects of RelA deficiency were observed at 1 h (Fig. 3e, clusters RA3 and RA5), whereas those of Rel deficiency were observed at 4 h and 18 h (Fig. 3f, clusters R1, R3 and R4). Absence of Rel also increased the inducible activation of many genes at 1 h (Fig. 3f, cluster R2). We conclude that RelA and Rel regulate temporally distinct phases of inducible gene expression in B cells.

**Rapidly activated genes.** Twenty-one of 113 RelA-binding genes in cluster I3 were shared with cluster RA3 (Fig. 4a and Extended Data Table 2). These genes were classified as being RelA-selective. Some of the best documented NF-κB target genes, such as *Tnfaip3*, *Gadd45b* and *Nfkbia*, were in this subset. This list also contained many genes, such as *Bhlhe40*, *Arl5b* and *Plaur*, that had not been previously associated

**Fig. 2 | Epigenomic characteristics of inducible RelA- and Rel-binding regions in activated B cells. a−d**, Chromatin accessibility and histone modifications were assessed by ATAC-seq and ChIP−seq, respectively, in splenic B cells activated with anti-IgM for 0, 1, 4 and 18 h. **a,b**, Pre-existing chromatin state in unactivated B cells at sites of inducible RelA binding at 1 h (left) and Rel binding at 18 h (right). Heatmaps were centered on summits of RelA or Rel peaks, ordered based on the strength of ATAC signal and segregated by genomic locations as indicated to the left of each panel. Boxed plots above each heatmap show the average profile across all sites at each genomic location (blue = promoters (defined as −1 kb to +0.1 kb relative to the transcription start site), yellow = intergenic regions and green = introns). **c**, Chromatin accessibility changes induced by activation were determined using DiffBind analysis of ATAC-seq data (fold change ≥1.5,

FDR ≤ 0.05). Transiently induced ATAC peaks (at 1 h) are shown on the top half (3,143 total peaks) and those induced at 18 h are shown at the bottom (2,261 total peaks), ordered according to the strength of the ATAC signal. H3K4me3 and RelA or Rel ChIP−seq profiles of the corresponding sites at the indicated time points are shown in adjacent panels. Boxed plots above each heatmap show the average ChIP−seq or ATAC-seq profile for each category of inducible ATC peaks (blue = 1 h induced ATAC peaks and green = 18 h induced ATAC peaks). Induced ATAC peaks at 1 h and 18 h were annotated to genomic regions shown on the right. **d**, Representative browser tracks of ATAC-seq peaks induced at 1 h (*Irf4*) or 18 h (*Dolk/Nup188*) and associated RelA or Rel binding at the indicated times. Transcriptional direction and transcription initiation sites are shown by red arrows. mm9 annotations are shown above the tracks.

with RelA. Many RelA-selective genes were upregulated in the absence of Rel (cluster R2), indicating that nuclear co-expression of both factors attenuated RelA-dependent early gene activation (Extended Data Table 2).

To circumvent possible developmental abnormalities in B cells from *RelA^{fl/fl}xCd19-cre* mice, we used Tat-Cre[25] to delete RelA in mature B cells (Supplementary Fig. 4b). As controls, WT (C57BL/6) B cells were treated identically. We activated control and RelA-depleted cells with

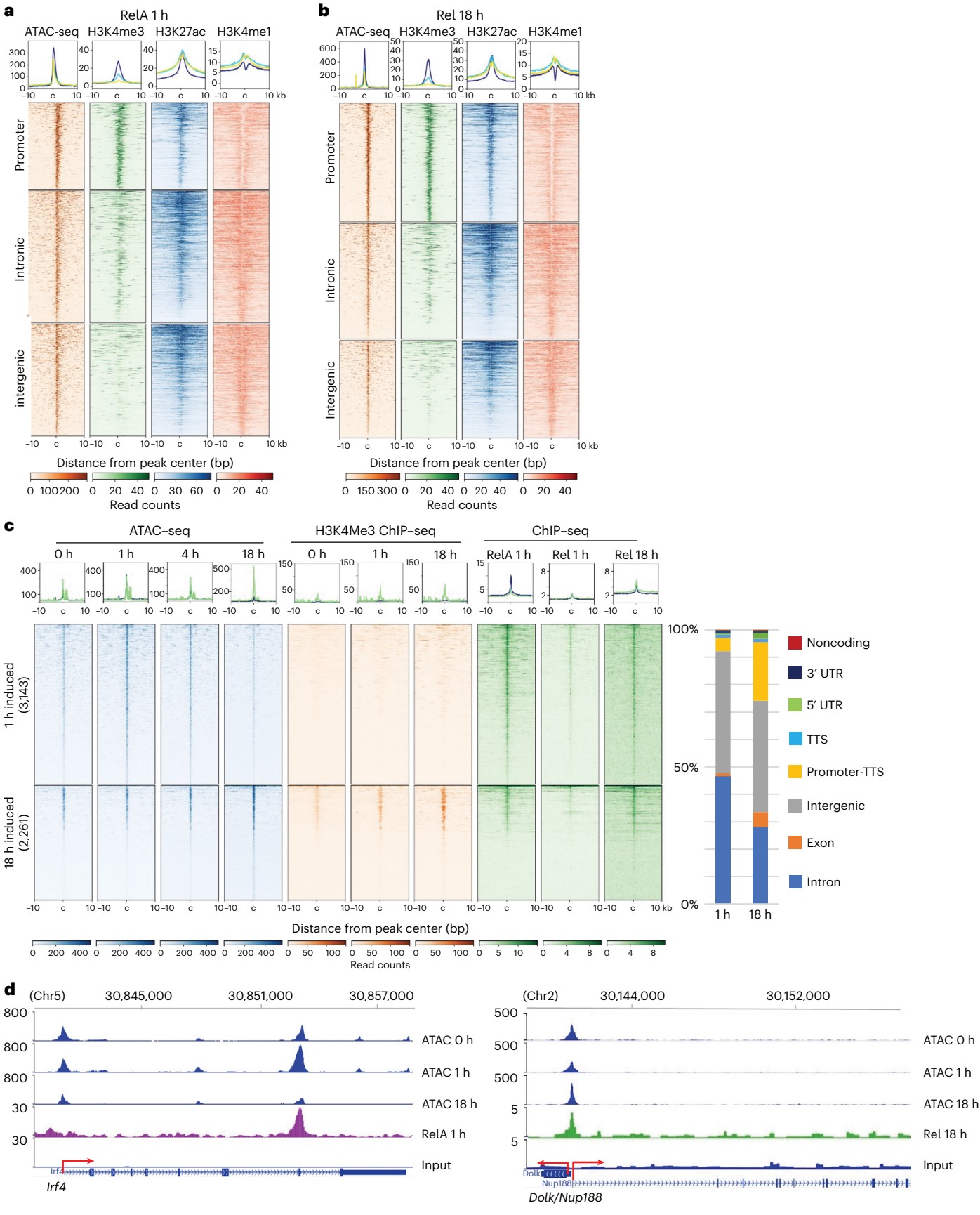

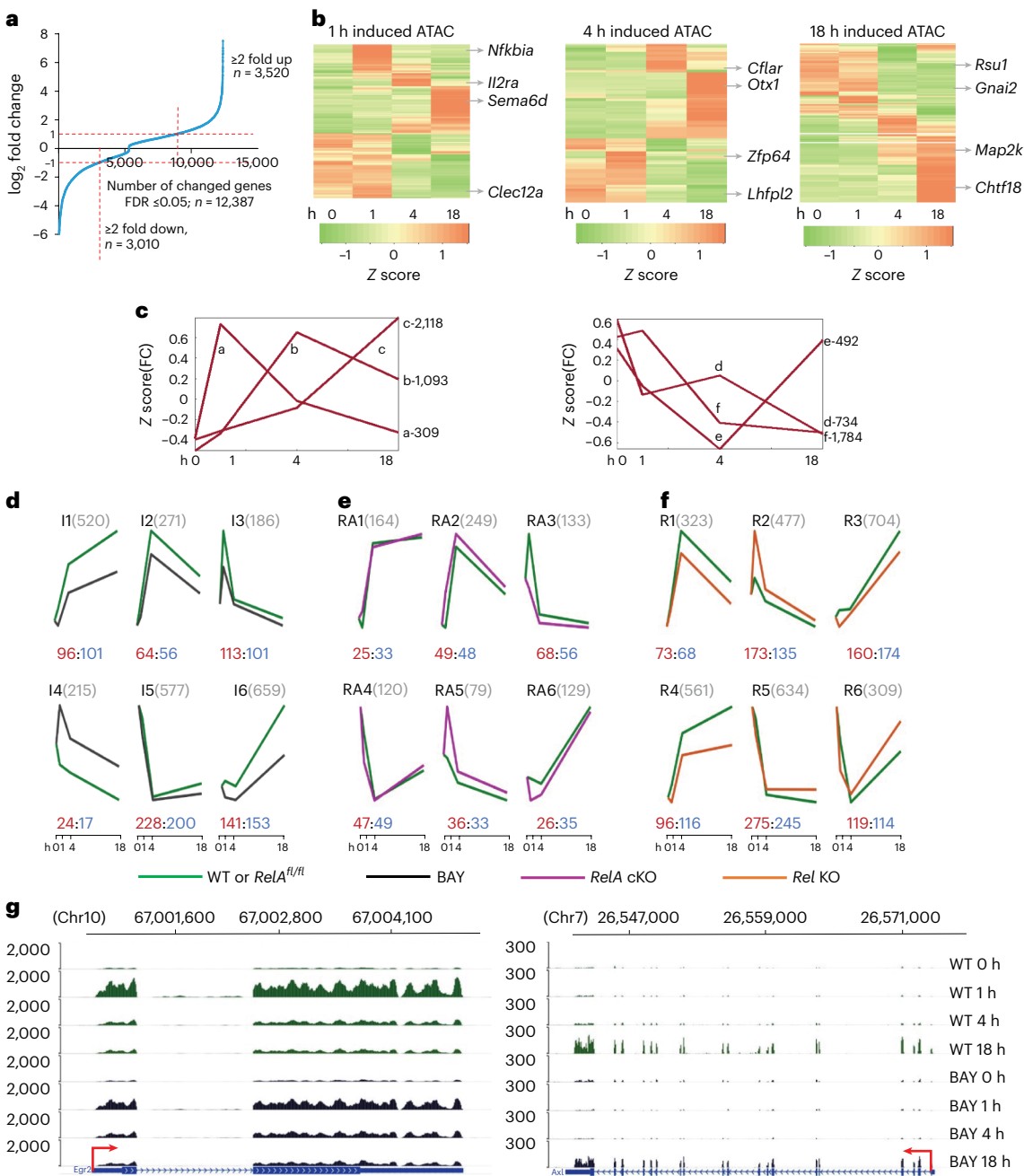

**Fig. 3 | NF-κB-dependent transcriptional responses in activated B cells. a–g,** Differential gene expression was analyzed using EBSeq (FDR ≤ 0.05) from two biological replicate experiments. **a,** Fold changes in gene expression at either 1, 4 or 18 h after activation compared to unactivated cells. Horizontal red dotted lines correspond to fold change threshold of ≥2. Numbers of genes upregulated or downregulated by ≥2× are indicated. **b,** Heatmaps represent expression levels of genes that were (1) differentially expressed by twofold at any time point with anti-IgM treatment and (2) associated with induced ATAC-seq peaks at either 1, 4 or 18 h time points as indicated above the heatmaps. Representative genes are shown on the right. **c,** Inducible transcripts were partitioned into three upregulated (patterns a–c) and three downregulated (patterns d–f) kinetic patterns by k-means clustering. Each graph represents the centroid profile of the average of z score of fold change (FC) compared with untreated cells at 1, 4 and 18 h. Total number of genes in each pattern are indicated on the right. **d–f,** RNA-seq analyses in activated B cells treated with IKK2 inhibitor BAY, or B cells that lack RelA or Rel. Differentially expressed genes were identified by DESeq2 (FDR ≤ 0.05). Genes that were induced at least twofold in control cells in each

group were further separated into six kinetic patterns by k-means clustering. Graphs represent the centroid profile of the average row z score of fold change for genes in each cluster color coded as described below. Names of clusters and numbers of genes in each cluster are shown on top. The numbers of genes to which RelA (1 h) and Rel (18 h) were bound are indicated in red and blue font, respectively, below the patterns. Activation times are noted on the x axis. **d,** Kinetic patterns of BAY-responsive transcripts in activated C57BL/6 WT B cells in the absence (green) or presence of BAY (black). **e,** Kinetic patterns of differentially expressed transcripts in activated B cells from control *RelA^fl/fl* (green) mice and *RelA* cKO mice (purple) activated with anti-IgM for times indicated on the x axis. **f,** Kinetic patterns of differentially expressed transcripts in activated B cells from control C57BL/6 mice (green) and *Rel* KO mice (orange) activated with anti-IgM for the indicated times. **g,** Representative browser tracks (mm9 annotation) from RNA-seq analyses of early (*Egr2*) and late (*Axl*) NF-κB target genes in WT and BAY-treated cells. Anti-IgM treatment times are shown on the right; transcriptional direction and transcription initiation sites are shown by red arrows.

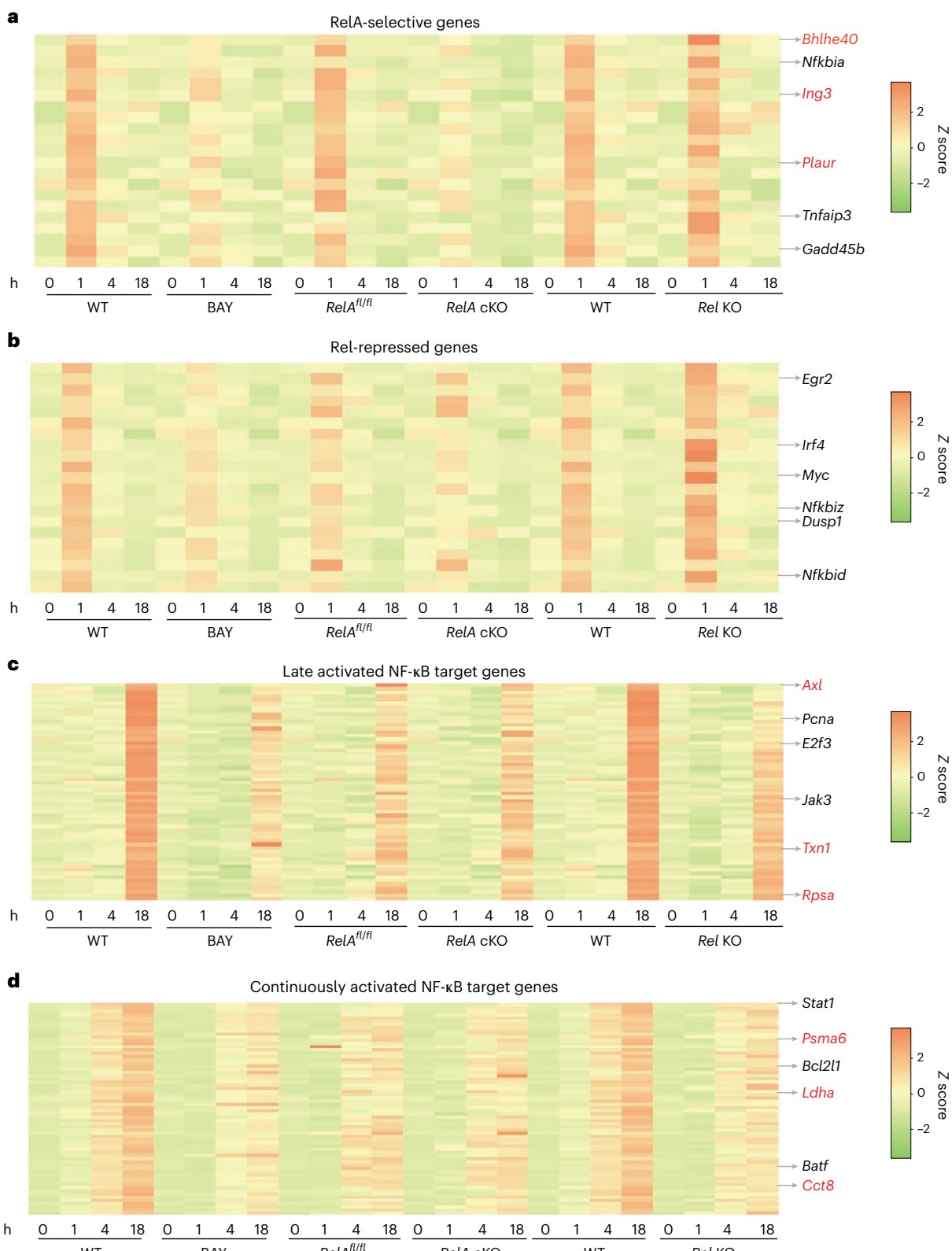

**Fig. 4 | Subunit selective expression patterns of NF-κB target genes.**
**a**–**d**, RNA-seq z score heatmaps of selected genes affected by BAY 11-7082 treatment (columns 1–8), RelA deficiency (columns 9–16) or Rel deficiency (columns 17–24). Each set of eight columns shows kinetic expression patterns in control (WT or *RelA*^fl/fl^) and test cells (BAY, *RelA* cKO and *Rel* KO) after 0, 1, 4 and 18 h of activation. Representative genes are noted on the right of each panel. Newly identified NF-κB target genes are noted in red font. **a**, Expression patterns of 20 RelA-selective genes defined as those whose induction was reduced by the inhibitor and RelA deficiency (overlap between clusters I3 and RA3 in Fig. 3d,e). Expression patterns of the same genes in Rel-deficient B cells are shown in the last eight columns (most of these genes fell in cluster R2 in Fig. 3f). **b**, Additional genes whose expression was increased in Rel-deficient compared to WT B cells (cluster R2 in Fig. 3f). **c**, Expression patterns of NF-κB target genes that were maximally induced between 4 h and 18 h of activation (cluster I6 in Fig. 3d). **d**, Those that were discernibly upregulated at 4 h but reached maximal expression 18 h (cluster I1 in Fig. 3d).

anti-IgM and evaluated the effects on RelA-selective gene induction. We found that activation profiles after *RelA* deletion in mature B cells closely resembled that of B cells from *RelA*^fl/fl^x*Cd19-cre* mice (Supplementary Fig. 4c).

Eighty-one of 113 NF-κB target genes in cluster I3 were not differentially regulated in RelA-deficient B cells (Extended Data Table 2), suggesting that they were activated by either RelA or Rel. Yet, many of these genes fell in cluster R2 indicating that RelA was a better activator compared to Rel (Fig. 4b). Our observations demonstrate differential activation potential of RelA or Rel on endogenous genes. This subset included genes such as *Dusp1*, *Nfkbid*, *Nfkbiz*, *Myc* and *Irf4*. A few genes in cluster I3 fell in RA5, indicating that they were positively regulated by Rel and suppressed by RelA at 1 h. We conclude that the simultaneous nuclear presence of both RelA and Rel sets up a state of mutual antagonism at early activation time points.

**Intermediate and late time points.** Most RelA/Rel-binding genes with maximal expression at 18 h (clusters I1 and I6) were unaffected by RelA deficiency and located in Rel-responsive clusters R3 and R4 (Figs. 3f and 4c,d and Extended Data Tables 3 and 4). We propose that these genes are direct targets of Rel. These included well-accepted Rel target genes (such as *Bcl2l1*), putative NF-κB target genes (such as *Icam1*) and many other genes that were not previously associated with Rel/NF-κB (such as *Axl*, *Ldha* and *Timm13*). Rel binding to a subset of these genes (Supplementary Fig. 4d) was further confirmed by ChIP–qPCR experiments (Supplementary Fig. 4e). Approximately one-third of NF-κB-responsive genes that were maximally expressed at 4 h (cluster I2) fell in cluster R1, marking them as Rel targets (Extended Data Table 5). We conclude that Rel-dependent gene expression occurs as early as 4 h after stimulation and dominates the NF-κB transcriptional landscape at longer time points. Most Rel target genes identified here have not been previously associated with NF-κB (Extended Data Tables 3–5).

To distinguish between NF-κB target gene expression in follicular and MZ B cells, we activated each cell type purified by fluorescence-activated cell sorting (FACS) (Supplementary Fig. 5a) with anti-IgM and carried out time-dependent RNA-seq. We found twofold fewer rapid, transiently induced genes in MZ cells (Supplementary Fig. 5b,c), and the NF-κB signaling pathway was missing in MZ cells at early time points (Supplementary Fig. 5d). Attenuated early NF-κB activation in MZ cells was substantiated by quantitative RT–PCR of select early induced NF-κB target genes (Supplementary Fig. 5e). These observations indicate that our present studies of NF-κB-dependent gene activation largely reflect responses of follicular B cells.

**Singe-cell analysis of NF-κB-dependent gene expression**
To further dissect NF-κB responses, we carried out scRNA-seq in activated B cells from WT, conditional RelA knockout (*RelA*^fl/fl^x*Cd19-cre*, *RelA* cKO) and Rel knockout (*Rel*^−/−^) mice. To focus on short- and long-term targets of RelA and Rel, respectively, we activated *RelA* cKO cells for 0, 1 and 4 h and *Rel*^−/−^ B cells for 0, 1 and 18 h. Uniform Manifold Approximation and Projection (UMAP) visualization showed relatively minor differences between genotypes, presumably reflecting alterations in only small subsets of genes due to the absence of specific NF-κB family members (Supplementary Fig. 6a). We focused on the behavior of RelA and Rel target genes identified by bulk RNA-seq analyses described above.

We observed markedly heterogenous responses of 20 RelA-selective genes (Fig. 5a). Some of the best-known NF-κB targets, such as *Tnfaip3*, *Gadd45b* and *Pim1*, were expressed in small proportions (5–25%) of activated cells. By contrast, *Rap1b* and *Rel* were expressed in relatively large proportions (70–90%) of cells. Single-cell analyses further corroborated reduced NF-κB activation in MZ B cells. Strongly RelA-dependent genes, such as *Nfkbia* and *Pim1*, were poorly induced in a cluster tentatively identified as MZ (based on the expression of *Cd1d1*, *Cd9*, *Tm6sf1*, *Dtx1*, *S1pr3* and *Cr2*) regardless of the RelA

genotype (Supplementary Fig. 6b). However, other early NF-κB target genes that are less RelA-dependent, such as *Samsn1* and *Bhlhe40*, were inducible in this subset of cells. Additional studies will be required to rigorously define the contribution of BCR-induced NF-κB signaling in MZ B cells. We quantified the effects of RelA deficiency by measuring the average scaled expression levels in control and RelA-deficient cells (Supplementary Fig. 6c). We found that kinetic patterns and RelA selectivity of genes identified by bulk RNA-seq were also evident at single-cell resolution.

To probe the basis for increased expression of many NF-κB target genes in *Rel*-deficient B cells, we first examined the status of RelA-selective genes in scRNA-seq from WT and Rel-deficient B cells. Many of these genes, such as *Arl5b*, *Nfkbia* and *Samsn1*, were upregulated on a per-cell basis in *Rel*^−/−^ B cells (Fig. 5b and Supplementary Fig. 6d). Increased expression in *Rel*^−/−^ B cells was also evident for genes that were not RelA-selective, including *Nfkbid*, *Trib2* and *Ptpn22* (Fig. 5c and Supplementary Fig. 6e). In addition to higher expression per cell, many of these genes were also expressed in a greater proportion of *Rel*^−/−^ B cells (for example, *Bhlhe40*, *Samsn1*, *Atf4* and *Irf4*; Fig. 5b,c). We conclude that nuclear co-expression of RelA and Rel attenuates RelA-dependent transcription. Reduced expression of *Rel* in Rel^−/−^ B cells may reflect positive autoregulation of the *Rel* gene promoter by Rel[26] or reduced stability of the noncoding transcript.

Rel target genes also showed substantial heterogeneity at the single-cell level (Fig. 5d and Supplementary Fig. 6f). Their responsiveness to Rel was evident in both reduced expression per cell and reduced proportions of cells expressing these genes at the 18 h time point (for example, *Timm13*, *Ldha* and *Bcl2l1*). A minority of genes showed the opposite trend in scRNA-seq from that predicted by bulk RNA-seq (for example, *Banf1*). Mechanisms underlying this pattern remain to be explored. Overall, single-cell studies strongly corroborated target gene predictions and kinetic patterns observed in bulk RNA-seq analyses and revealed wide variations in NF-κB responses in BCR-activated B cells.

**NF-κB-initiated cascades**
Many genes that were activated in B cells and inhibited by BAY did not bind RelA or Rel in ChIP–seq assays (Fig. 6a and Supplementary Fig. 7). We reasoned that these genes were targets of transcription factors activated by NF-κB and referred to as indirect targets of NF-κB (ref. 19). Of the many transcription factor genes induced in response to BCR signaling (Fig. 6b), a subset met our criteria for being direct NF-κB targets (Extended Data Table 6). Early activated transcription factor genes included previously known NF-κB targets such as *Fos*/*Fosb*, *Myc* and *Irf4* (http://www.bu.edu/NF-κB/gene-resources/target-genes/), as well as new genes such as *Tgif1/2*, *Bhlhe40* and *Atf4* (Fig. 6c). Late-induced NF-κB-responsive transcription factor genes were targets of Rel, and most had not been previously associated with NF-κB (Fig. 6d and Extended Data Table 6).

To test the idea that indirect NF-κB target genes were activated by NF-κB-induced transcription factors, we searched for DNA-binding motifs in promoters (−400 bp to +100 bp relative to transcription start site) of genes that were upregulated at 4 h (clusters I2 + I1) and 18 h (clusters I1 + I6). Interferon-stimulated regulatory elements and Myc-binding sites were enriched in promoters of genes induced at 4 h (Fig. 6e), suggesting that a subset of indirect NF-κB target genes were direct targets of Irf4 and Myc. Promoters of indirect target genes that reached the highest levels of expression at 18 h were enriched for binding sites NF-Y and E2F4 (Fig. 6f). Rel targets included the gene encoding the closely related factor *E2f3* (Fig. 6d). The basis for the emergence of the NF-Y motif among late activated indirect target genes remains unclear. We hypothesize that Rel-regulated transcription factors, such as Arid3a, Bcl11a, Rxra, Bcl6b, Hhex and several zinc finger proteins, whose motifs were not identified in this analysis, activate gene expression from promoter distal sequences. We conclude that early and late NF-κB induce an array of transcription factor genes that activate

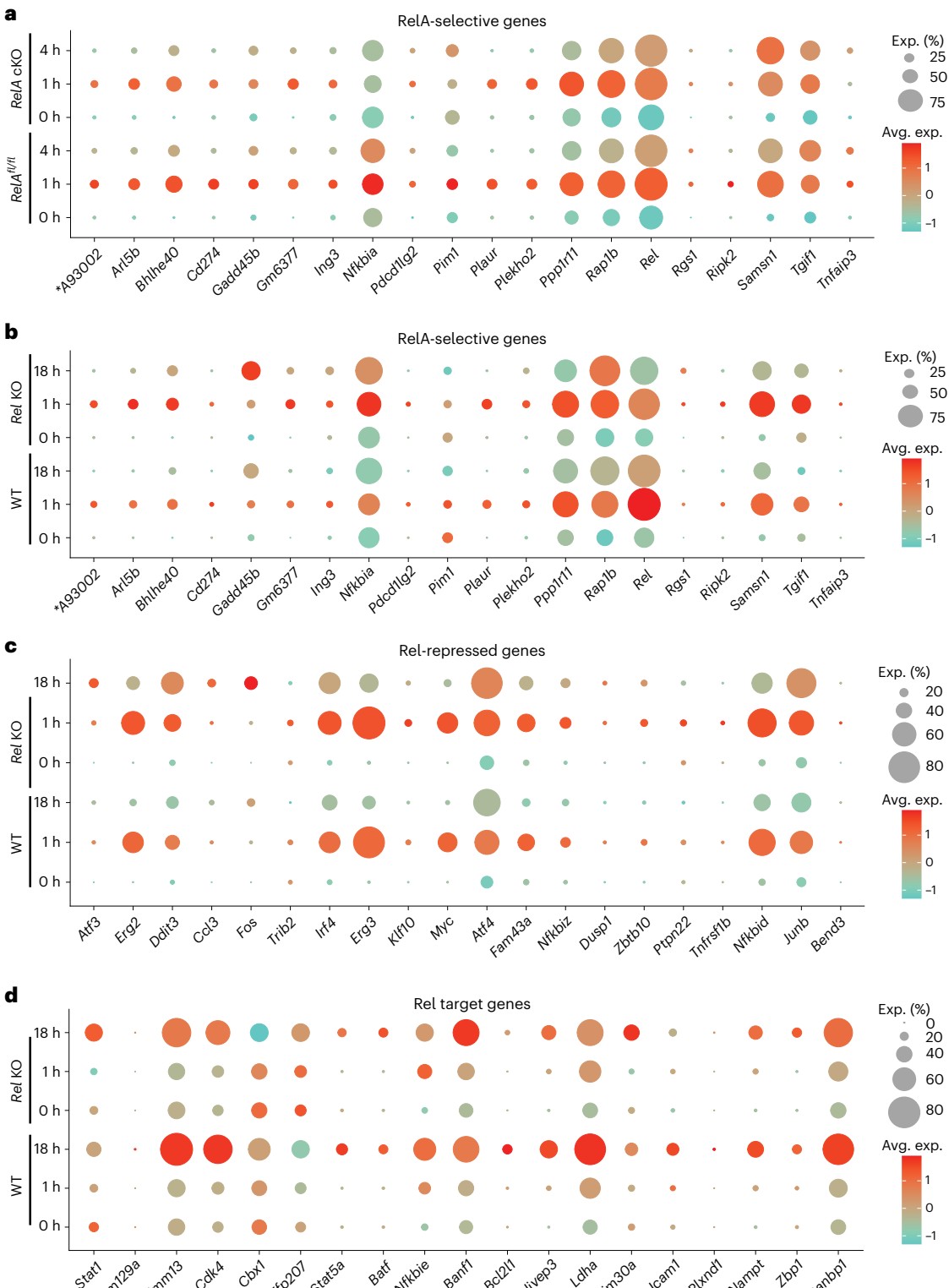

**Fig. 5 | scRNA-seq analysis of NF-κB target genes in activated B cells.**
**a–d**, scRNA-seq was carried out with anti-IgM-activated B cells from *RelA^fl/fl*
(control) and *RelA^fl/fl*x*Cd19-cre* (*RelA* cKO) for 0, 1 and 4 h or from C57BL/6 (WT)
and *Rel^−/−* B cells treated with anti-IgM for 0, 1 and 18 h. In total, 10,000 B cells
were assayed from each condition. Data analysis was carried out by Seurat as
described in Methods. Dotplot visualizations of differential gene expression
patterns of select NF-κB target genes are shown. Gene expression changes are
plotted by both scaled average expression (avg. exp., illustrated by dot color)
and the percentage of cells found to express that gene (exp (%), depicted by
the size of the dot). Cellular origin of libraries is indicated on the *y* axis.

**a**, Single-cell analysis of RelA-selective genes from Fig. 4a (*x* axis). Expression
levels and proportions of gene-expressing cells in control and *RelA* cKO B cells are
depicted at 0, 1 and 4 h. **b**, Expression patterns of RelA-selective genes from
Fig. 4a are depicted in WT and Rel-deficient B cells activated for 0, 1 and 18 h.
**c**, Single-cell expression patterns of additional Rel-repressed genes (from
cluster R2 in Fig. 3f) in WT and Rel-deficient B cells activated for 0, 1 and 18 h.
**d**, Single-cell analysis of Rel target genes (from clusters R3 and R4 in Fig. 3f) in
WT and Rel-deficient B cells activated for 0, 1 and 18 h. Additional quantitation
for all patterns is shown in Supplementary Fig. 6c–f. An asterisk indicates that
A93002 = A930024E05Rik.

cascades of NF-κB-regulated gene expression in activated B cells. Both direct and indirect NF-κB target genes were implicated in oncogenic processes in the CancerSea database (Supplementary Fig. 8a,b).

## Discussion

Our studies reveal that two closely related NF-κB family members drive kinetically distinct patterns of gene expression in activated B cells. Rigorous identification of many previously unknown targets of each factor revealed transcriptional cascades initiated by NF-κB activation that impact B cell function and cancer. We found that both RelA and Rel contributed to early transient gene expression. Simultaneous nuclear presence of RelA and Rel resulted in attenuated RelA-dependent gene expression that was manifested both at the expression level per cell and in proportions of cells that activated specific target genes. By contrast, late NF-κB target genes were largely Rel-dependent. We also noted that NF-κB transcriptional responses were markedly heterogeneous across individual cells. Indeed, some of the best-known target genes, such as *Tnfaip3* and *Gadd45b* among early genes and *Bcl2l1* among late genes, were induced in less than 20% of cells. Implications of our findings for B cell biology and cancer follow.

Genome-wide RelA and Rel binding was highest at times of maximal nuclear expression of each factor, at 1 h and 18 h postactivation, respectively. Residual RelA binding at 4 h occurred largely at sites that were also RelA-bound at 1 h. Most Rel binding at 1 h overlapped with RelA, setting up competition for binding at a subset of sites. Bulk and scRNA-seq analyses showed that nuclear co-expression of Rel and RelA down modulated RelA-dependent gene expression. Rel has been previously proposed to repress select inflammatory genes in TNFα-activated fibroblasts by recruiting HDAC1 (ref. 27). We have not investigated whether this mechanism applies in BCR-activated B cells. However, we favor the idea that repressive functions of Rel result from competition for binding sites, with Rel being a poorer transcriptional activator compared to RelA. About 40% of sites that bound RelA at 1 h also bound Rel at 18 h. In principle, early RelA binding followed by late Rel binding could confer persistent NF-κB activity as has been noted for sequential RelA and RelB recruitment[28]. This proved not to be the case.

It is interesting to note that approximately 2,000 sites were exclusively occupied by either RelA (at 1 h) or Rel (at 18 h; note that the use of 'exclusive' here is constrained by thresholds used to define ChIP–seq peaks during analysis). This discrimination was not based on sequences underlying RelA- or Rel-specific peaks because both sets were enriched for Sfpi1 and NF-κB motifs. Nor was it governed by chromatin accessibility because both regions were premarked with active chromatin marks in resting B cells. One possibility is that levels of post-translationally induced Rel at 1 h may be lower than de novo translated Rel present at 18 h, leading to binding site selectivity by mass action. Alternatively, site selectivity may be determined by subunit composition. Pre-existing RelA or Rel that translocate early have been proposed to largely consist of p50-containing heterodimers[29], whereas higher levels of Rel at later time points may have substantial proportions of Rel homodimers.

Among 113 transiently induced NF-κB target genes that bound RelA at 1 h, only 21 were adversely affected in RelA-deficient B cells. We infer that Rel does not effectively substitute for the absence of RelA at these genes

and termed them as being RelA-selective. The relatively few RelA-selective genes identified in this study may be a special feature of activated B cells that express robust levels of Rel. In cells that express lower levels of Rel, RelA-containing NF-κB may be the dominant factor activating these genes. Most of the 113 early NF-κB target genes also bound Rel at 18 h, yet RNA levels peaked at 1 h. Thus, sequential binding of RelA and Rel did not confer protracted transcriptional activity of these genes.

We identified an additional 254 NF-κB target genes that reached maximal expression at late time points (clusters I1 and I6). These genes were negatively affected by the absence of Rel but not RelA, marking them as Rel targets. Approximately 60% of late target genes (cluster I6) bound RelA at 1 h, yet maximal RNA levels occurred coincident with Rel binding at 18 h. Despite RelA binding at 1 h, expression of these genes was unaltered by RelA deficiency, indicating that early RelA binding was not required for their maximal expression at later times. These observations demonstrate that RelA or Rel binding per se is not an accurate measure of transcriptional activity induced by these factors.

We hypothesize that the late transcriptional activity of Rel target genes is determined by the nature of neighboring sequences. We found that RelA-bound regions associated with the 113 early and transiently induced NF-κB target genes (cluster I3) were enriched for NF-κB and Sfpi1 binding motifs, whereas Rel peaks associated with 153 late activated NF-κB-responsive genes (cluster I6) were enriched for NF-κB and IRF binding motifs. Because several *Irf* genes were induced in response to BCR stimulation, this pattern is consistent with the notion that late Rel works with factors that are upregulated at earlier stages. Interestingly, some *Irf* genes are themselves early NF-κB targets, suggesting that early NF-κB target genes may cooperate with late-induced Rel to activate late gene expression.

Indirect NF-κB targets were affected by combined blockade of RelA and Rel but did not bind either protein. Our working hypothesis is that the majority of these genes are regulated by NF-κB-induced transcriptional regulators. Indeed, many genes encoding transcription factors were among the few hundred direct NF-κB targets we identified. Some are well-established, such as *Myc* and *Irf4*. Many more were newly identified targets, including *Tgif1* and *Tgif2* that are implicated in different kinds of cancers[30–34], stress-responsive transcription factors (*Atf3* and *Atf4*) that are implicated in crosstalk with NF-κB in tumors[35–38] and *Bhlhe40*, a transcription factor implicated in immune function[39].

The group of new Rel-regulated transcription factors was also biologically meaningful. Two new direct targets included *Hhex* and *Bcl6b*, both of which have been shown to be involved in optimal GC responses[40–42]. Although GC responses are severely reduced in germline *Rel* knockout mice[14,24], Rel target genes that manifest their critical role in GCs are not known. Our evidence that Rel directly activates two key GC factors provides the first insight into pathways by which Rel impacts GC responses. Another Rel target identified here, *Bcl11a*, also links Rel to the GC. *Bcl11a* has been shown to be highly expressed in GCs[18,43], but its functional role has not been evaluated. It is not our intention to claim that 18 h activated B cells are equivalent to GC B cells. We surmise that connections between Rel and *Hhex/Bcl6b/Bcl11a* established here may be strengthened in the GC microenvironment to optimize Rel-dependent responses.

**Fig. 6 | NF-κB-activated transcriptional cascades. a**–**f**, Genes whose expression was altered by blocking both RelA and Rel induction with BAY 11-7082 but did not bind either RelA or Rel were inferred to be targets of NF-κB-induced transcription factors. **a**, Such genes were identified in four of the six *k*-means clusters in BAY-treated cells (Fig. 3d). Graphs show average row *z* score of fold change for genes in each cluster in activated WT B cells in the absence (green) or presence of BAY (black; from Fig. 3d). Activation times are noted on the *x* axis. Numbers on the top of each plotted kinetic pattern denote total number of genes in that cluster; numbers below each plot denote the numbers of genes that did not score for RelA or Rel binding at any activation time by ChIP–seq analyses. **b**, Heatmap profiles of three transcription factor gene families induced in response to B cell activation. Representative NF-κB target genes are indicated to the right of each panel (red font). See Extended Data Table 6 for a complete list of NF-κB target genes that encode transcription factors. **c**,**d**, Browser tracks of early (**c**) and late (**d**) transcription factor genes that were affected by BAY treatment. The top eight lines show RNA-seq tracks at different time points of anti-IgM treatment in the presence (BAY) or absence (WT) of BAY. The bottom two lines show RelA 1 h or Rel 18 h ChIP–seq and associated input tracks. **e**,**f**, DNA-binding motifs enriched in promoters (−400 to +100 bp relative to the transcription start site) of indirect NF-κB targets at 4 h (**e**) and 18 h (**f**). Motifs were identified using the known motifs setting in the HOMER algorithm. Significance was calculated using default statistical setting provided by HOMER.

NF-κB has been shown to be necessary for a subset of human B cell lymphomas[44,45]; translocations of *REL* and *NFKB2* have been noted in others[16,46]. Target genes that mediate NF-κB's oncogenic potential are not well characterized. Both early and late target genes identified here provide insights into this question. Early target genes were linked to hypoxia, inflammation and quiescence in the CancerSea database,

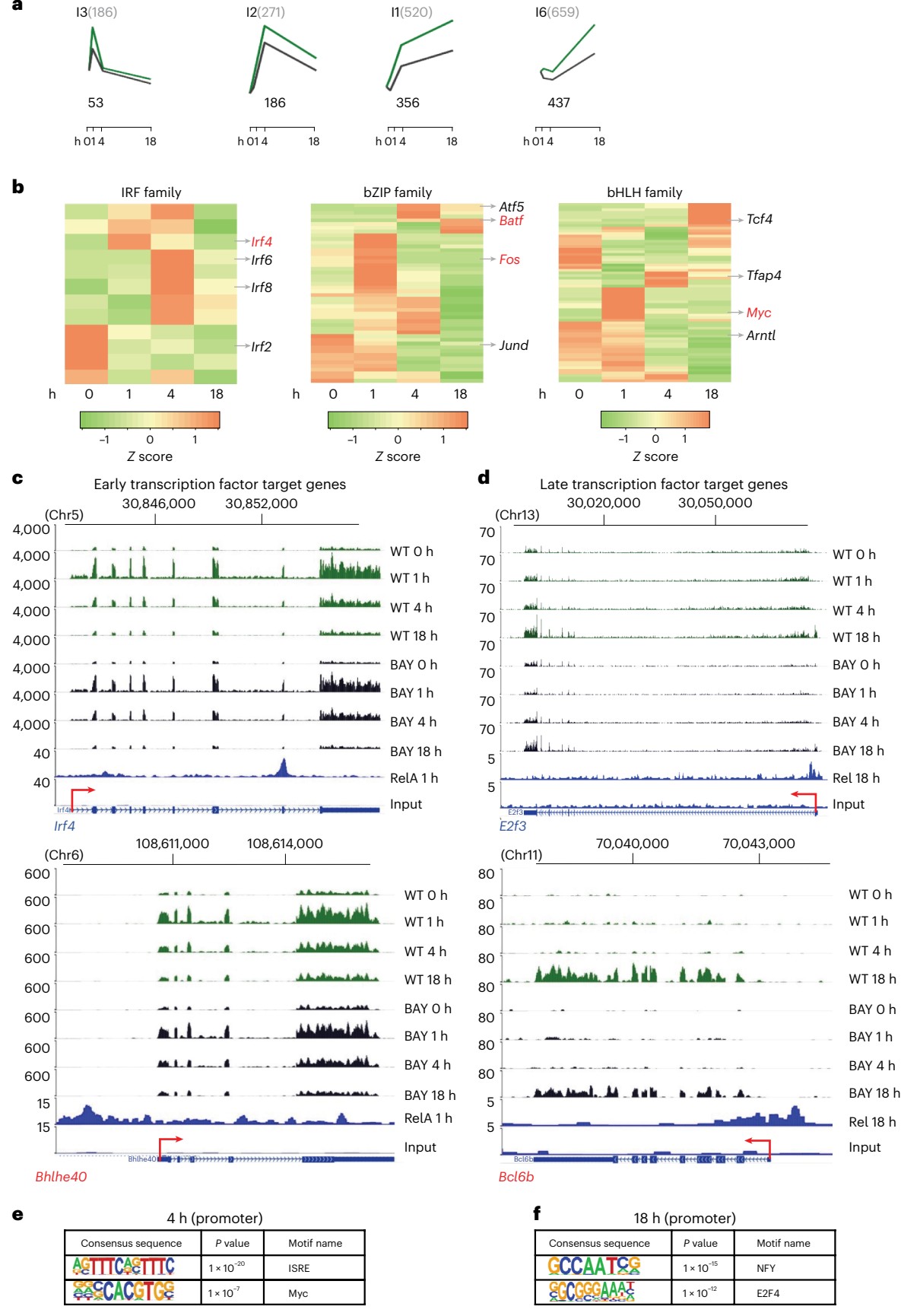

suggesting a role for NF-κB in initiating oncogenesis. By contrast, late target genes were associated with cell cycle, DNA damage and repair and invasion, suggesting continued impact of NF-κB during cancer progression. Several genes in our list of Rel targets, such as *Arid3a*, *Bcl11a* and *Bcl6b*, could serve as downstream effectors of ectopic Rel activation during lymphomagenesis[47]. Additionally, the Rel target gene *Maz* has been implicated in cancers[48,49] and as a multiple sclerosis susceptibility gene[50]. Most early induced NF-κB target genes associated with cancers (such as *Tgif1,2* and *Atf3,4*) were also Rel-responsive, indicating that early and late Rel target genes may drive oncogenesis. Because many of these target genes encode transcription factors, NF-κB-initiated transcriptional cascades may further amplify its oncogenic functions.

## Online content

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

## Methods

### Mice, B cell preparation, cell culture and antibodies

Primary B cells were purified from the spleens of 6- to 12-week-old WT (C57BL/6), *RelA*[fl/fl] (ref. [51]), RelA cKO (*RelA*[fl/fl]x*Cd19*-cre; generated by crossing *RelA*[fl/fl] with *Cd19-cre* (Jackson Laboratory, 006785)) and *Rel* KO (*Rel*[−/−]) (ref. [17]). For each experiment, five to six age-matched mice (mixed male/female) were used. Additional randomization was not required for these studies because the data were obtained ex vivo from pooled B cells obtained from multiple mice. Data collection and analysis were not performed blind to the conditions of the experiments. No animals or data points were excluded from the analyses for any reason.

All studies were conducted under the guidance of the Institutional Animal Welfare Assurance Number−NIH Intramural Research Program, D16-00602; NIA AAALAC unit, 000401. Mice were housed in an intramural NIH vivarium with full AAALAC International accreditation. The circadian light cycle is 12 h light and 12 h dark that are on automatic timers with electronic monitoring monitored by the Siemens building control system. The temperature is set at 72 °F (range, 69−75 °F) with a humidity range of 30−70%. The mice are fed an ad libitum diet from Envigo (Envigo 2018SX Global 18% protein extruded rodent diet). The Baltimore municipal water is filtered by reverse osmosis and then chlorinated at 2−3 ppm. All housing standards are as per the Guide for the Care and Use of Laboratory Animals.

Single-cell suspensions from the spleen were prepared in RoboSep buffer (Stemcell Technologies, 20104) and enriched for B cells by negative selection using EasySep Mouse B cell Isolation Kit (Stemcell Technologies, 19854). B cells were cultured at $2 \times 10^6$ cells per ml in RPMI 1640 medium supplemented with 10% heat-inactivated FBS (HyClone, SH30088.03), penicillin−streptomycin−glutamine (Invitrogen, 10378016) and 55 nM 2-mercaptoethanol (Sigma-Aldrich, M6250). Follicular and MZ B cells were FACS sorted, as previously described[52], using BD FACS-Aria Fusion. Follicular B cells were sorted as B220+ (BioLegend, 103240), AA4.1− (BioLegend, 136510), CD21lo (BioLegend, 123410) and CD23hi (BioLegend, 101606). MZ B cells were sorted as B220+, AA4.1− and CD21hiCD23lo. TAT-Cre treatment was as follows: splenic B cells ($10 \times 10^6$ ml[−1]) from *RelA*[fl/fl] and C57BL6 (WT, as a control) were treated with 100 µg of TAT-Cre (Excellgen, EG-1001) for 45 min at 37 °C, in humidified CO$_2$ incubator and then washed. Using a dead cell removal kit (Miltenyi Biotec, 130-090-101), live cells were isolated after TAT-Cre treatment and recultured for 72 h in RPMI 10% FBS supplemented with 50 ng ml[−1] of BAFF (R&D Systems, 2149-BF). After 72 h, cultures were washed using warm RPMI, and cells were cultured at $2 \times 10^6$ cells per ml until the next steps. Cells were activated with 10 µg ml[−1] goat anti-mouse IgM Fab'2 (Jackson Laboratory, 115-006-075) for 1, 4 and 18 h. After activation, dead cells were removed with dead cell removal magnetic microbeads (Miltenyi Biotec, 130-090-101). To inhibit IKK signaling, cells were pretreated with 1 µM BAY 11-7082 (Calbiochem, B5556) for 1 h before anti-mouse IgM stimulation. The following antibodies were used for ChIP, ChIP−seq or western blot: anti-RelA (Santa Cruz Biotechnology, sc-372), anti-Rel (Santa Cruz Biotechnology, sc-71), anti-hnRNP A1 (Santa Cruz Biotechnology, sc-32301), anti-H3K27ac (Active Motif, 39133), anti-H3K4me1 (Abcam, ab8895) and anti-H3K4me3 (Abcam, ab8580). Horseradish peroxidase-coupled goat anti-mouse IgG (Santa Cruz Biotechnology, sc-2005) and goat anti-rabbit IgG (Santa Cruz Biotechnology, sc-2004) were used for immunoblotting.

### Immunoblotting

Proteins were separated by electrophoresis through 10% SDS−PAGE and electrophoretically transferred to nitrocellulose membrane. After blocking with 5% dried milk in Tris−HCl (pH 7.4)-buffered saline/0.05% Tween (TBST) for 1 h, membranes were incubated with primary antibodies overnight at 4 °C, washed in TBST and incubated for 1 h with horseradish peroxidase-coupled secondary antibodies. Proteins were detected by using the enhanced chemiluminescence systems (Pierce Biotechnology, 32106) and imaged using Syngene Imaging System.

### Flow cytometry and image cytometry

Isolated B cells ($0.5 \times 10^6$ cells) from spleens were fixed with 2% paraformaldehyde (Electron Microscopy, 15713) for 10 min in the dark at room temperature (23 °C). Fixed cells were washed with PBS and resuspended in ice-cold Perm Buffer III (BD Biosciences, 558050) and incubated overnight at −20 °C. Cells were washed with PBS and resuspended in BD perm/wash buffer (BD Biosciences, 554723). Cells were either stained with rabbit-anti mouse RelA (Cell Signaling Technology, 8242S; 1:1,500) or isotype (Cell Signaling Technology, 3900; 1:1,000) antibody for 1 h at 4 °C. Cells were washed with perm/wash buffer and restained with anti-rabbit Fab'2-AF647 antibody (Cell Signaling Technology, 4414; 1:1,000) for 30 min. Similarly activated cells were stained with the following antibodies: Myc-AF-647 (Cell Signaling Technology, 13871; 1:50), Egr2-APC (eBiosciences, 17-6691-82; 1:50), Rel-PE (eBiosciences, 12-6111-80; 1:200), Ikba-PE (Cell Signaling Technology, 7523; 1:100), Dec1-AF-594 (Novus Biologicals, 1800AF594; 1:100), Ezh2-PE (BD Biosciences, 562478; 1:50), CD72-BV421 (BD Biosciences, 740058; 1:200) and HSP90-β (Invitrogen, PA3-012; 1:1,500). HSP90-β-stained cells were further stained with anti-rabbit Fab'2-PE antibody (Cell Signaling Technology, 79408; 1:1,000) for 30 min. Finally, cells were washed and acquired on BD Aria Fusion or BD Symphony flow cytometer, and data were analyzed with FlowJo v10.1.

For image cytometry, cells were stained as above for RelA after anti-IgM (F(ab)'2) or PMA (Sigma-Aldrich, P1585) and Ionomycin (Sigma-Aldrich, I0634) treatment and later resuspended in perm/wash buffer solution containing DAPI (500 ng ml[−1]; Thermo Fisher Scientific, 62248) and RNase A (1 µg ml[−1]; Invitrogen, 912091021). Stained cells were acquired on Amnis ImageStream X Mk II Imaging Flow Cytometer (Luminex), and data were analyzed using IDEAS 6.0 software (Luminex).

### ChIP and ChIP−seq library construction

All ChIP experiments ($n = 2$) were performed as previously described[19]. Briefly, for H3K27ac and H3K4me3 ChIP, cells were washed twice with PBS, fixed at room temperature (23 °C) with 1% formaldehyde in PBS for 10 min. For RelA and Rel ChIP, cells were crosslinked with 1.5 mM EGS (Pierce Biotechnology, 21565) for 30 min followed by 1% formaldehyde (Sigma-Aldrich, F8775) at room temperature (23 °C) for 15 min in PBS. Reactions were quenched by adding glycine to a final concentration of 0.125 M, and cells were washed twice with cold PBS. Cells were resuspended in cell lysis buffer (50 mM HEPES−KOH (pH 7.5), 150 mM NaCl, 1 mM EDTA, 1% Triton X-100, 0.1% Na-deoxycholate and 0.1% SDS), rotated for 15 min at 4 °C and then centrifuged at 1,200g for 10 min at 4 °C. The nuclear pellet was resuspended in nuclear lysis buffer (50 mM HEPES−KOH (pH 7.5), 150 mM NaCl, 1 mM EDTA, 1% Triton X-100, 0.1% Na-deoxycholate and 1% SDS) and rotated for 15 min at 4 °C. After centrifugation, the pellet was lysed in buffer containing 50 mM HEPES− KOH (pH 7.5), 150 mM NaCl, 1 mM EDTA, 1% Triton X-100, 0.1% sodium deoxycholate, 0.1% SDS and protease inhibitors. The crosslinked chromatin was fragmented by sonication (Branson sonicator) to a shear size of 200−2,000 bp. ChIP was performed with 2.5 µg anti-RelA or anti-Rel antibody, 3 µg anti-H3K27ac or anti-H3K4me3 antibody prebound to 50 µl Protein A (Invitrogen, 10001D) or G Dynabeads (Invitrogen, 10003D). Sonicated chromatin was added to antibody-bound beads and incubated at 4 °C overnight. Beads were collected by centrifugation, washed and incubated at 65 °C for 4 h in elution buffer (50 mM Tris−HCl (pH 7.5), 10 mM EDTA and 1% SDS) to reverse crosslinking. ChIP DNA was purified by phenol−chloroform extraction followed by ethanol precipitation. ChIP−qPCR was performed using Sybr Green Fast (Applied Biosystems, 4385610). The *y* axis for Fig. 1c was calculated by the following formula: fold change = $2^{(CT(input)−CT(target))}/2^{(CT(input)−CT(neg control))}$. The *y* axis for Supplementary Fig. 4e represents ChIP signals at target amplicons compared to input control. Primer sequences are provided in Supplementary Table 1.

For sequencing, adapters were ligated to the precipitated DNA fragments or the input DNA to construct a sequencing library according to the

manufacturer's protocol for the NuGEN Ovation Ultralow DNA-seq library preparation kit (0344-32, Tecan). A 75 bp single-end read sequencing was performed on Illumina NextSeq 500 instrument. Two biological replicate ChIP–seq experiments were carried out for each antibody.

## ATAC-seq library construction

ATAC-seq library construction ($n = 2$) was performed as previously described with minor modifications[53]. In total, 100,000 B cells were lysed with 100 µl of lysis buffer (10 mM Tris–HCl (pH 7.4), 10 mM NaCl, 3 mM MgCl$_2$, 0.1% NP-40 and 0.1% Tween-20). After centrifuging at 500x $g$ for 10 min at 4 °C, pelleted nuclei were resuspended with 50 µl of transposition mix (FC-121-1030, Illumina; 1× Tagment DNA buffer, Tn5 Transposase, nuclease-free H$_2$O) and incubated for 30 min at 37 °C in a thermomixer. Transposed DNA was purified using MinElute columns (Qiagen, 28004) and subsequently amplified with Nextera sequencing primers and NEB high fidelity 2× PCR master mix for 12 cycles (New England Biolabs, M0541). PCR-amplified DNA was purified using MinElute columns and sequenced using the Illumina NextSeq sequencer with paired-end reads of 150 bases.

## RNA isolation for quantitative RT–PCR and RNA-seq of bulk B cells and Fo/MZ cells

From resting or activated B cells, follicular and MZ B cells, total RNA ($n = 2$) was isolated using the RNeasy Plus Mini Kit (Qiagen, 74134). For total B cells, cDNA was synthesized with the SuperScript First-Strand Synthesis System (Invitrogen, 11904018), while for follicular and MZ B cells, cDNA was synthesized with SuperScript IV VILO Master Mix (Invitrogen, 11756050). PCR was performed in duplicate using the ABI ViiA 7 Real-Time PCR System using QuantStudio 1.6.1 (Thermo Fisher Scientific, 4453545) with iTaq Universal SYBR Green Supermix (BioRad, 1725150). The primers are listed in Supplementary Table 2.

For RNA-seq, total RNA from B cells was used for barcoded library preparation using Illumina TruSeq total RNA preparation kit (Illumina, 20040525). Samples were sequenced at the Johns Hopkins Transcriptomics and Deep Sequencing Core using the Illumina HiSeq 2000 with 75 bp single-end reads.

For follicular and MZ B cells, indexed libraries were generated from 10 ng total RNA from two biological replicates using the SMARTer Stranded Total RNA-seq Kit v2—Pico Input Mammalian (Takara Bio, 634412) per manufacturer's protocol. The libraries were sequenced on an Illumina NovaSeq 6000 using 100 bp paired-end reads. BCL files were demultiplexed and converted to FASTQ files using Illumina's bcl2fastq program (v2.20.0.422).

## RNA-seq analysis

RSEM[54] (v1.3.2) was used to align RNA-seq reads to the mouse genome and quantify transcript abundance. EBSeq[22] (v1.24.0) was used to produce normalized reads for all samples and to identify differential RNA expression in WT, follicular and MZ samples across the entire time course. Comparisons between WT and *Rel* KO or *RelA* cKO or BAY 11-7082-treated samples were carried out in DESeq2 (ref. [55]). Differentially expressed genes were identified using DESeq2 (v1.22.1) following the model: ddsTC <- DESeqDataSetFromMatrix(countData = countData, colData = colData, design = ~Time+Treat+Time:Treat; ddsTC <- DESeq(ddsTC, test="LRT", reduced = ~ Treat+Time). Two filtering steps were followed to define differentially expressed genes in cells with modified NF-κB expression. First, all differentially changed genes were selected based on an FDR ≤ 0.05 after DESeq2 analysis; second, only those genes were selected that were more than twofold upregulated or downregulated in wild samples after anti-IgM treatment. *K*-means analysis of RNA expression data was carried out in MATLAB using fold change compared with 0 h, with correlation as the distance metric, repeat clustering set to five and other parameters set to default. RNA-seq analyses were performed with activated B cells treated with IKK2 inhibitor BAY or B cells that lack RelA or Rel. Genes that were

induced at least twofold in control cells in each group were further separated into six kinetic patterns by *k*-means clustering. Names of clusters and numbers of genes in each cluster are shown in Fig. 3d–f. A number of genes to which RelA (1 h) and Rel (18 h) binding were annotated are indicated in red and blue font, respectively, below the patterns. Activation times are noted on the *x* axis. For visualization, bigwig files were generated. *RelA*$^{fl/fl}$ and *RelA* cKO samples were aligned to mouse genome assembly mm10, all other bigwig files were aligned to mm9. All samples were normalized to 1 million aligned reads for visualization. CancerSEA[56] analysis in Supplementary Fig. 8a,b was analyzed in the link http://biocc.hrbmu.edu.cn/CancerSEA/ with default parameters set. Enriched KEGG pathways from the genes were identified using the web server tool ShinyGO 0.77 (http://bioinformatics.sdstate.edu/go/)[57].

## Analysis of ChIP–seq data

Bowtie1 (v1.0) software was used to map RelA and Rel quality-filtered reads from demultiplexed FASTQ files to mouse genome assembly mm9 (Ensembl v67) with default options. CisGenome software (http://www.biostat.jhsph.edu/~hji/cisgenome/) was used for peak calling with a window size of 200 bases and a threshold of four reads (two from each strand), FDR ≤ 0.1 and fold change ≥ 2. Peaks separated by less than 200 bases were merged into one peak. For RelA ChIP–seq peak calls, we selected the peaks that were more than tenfold changed compared to input, and for Rel peak calls, we selected the peaks that are more than fivefold changed compared to input in the Rel ChIP–seq. For histone ChIP–seq, we used Bowtie2 (v2.9.2) (ref. [58]) with default parameters, aligned on the mouse genome assembly mm9. We used HOMER annotatePeaks.pl to annotate peaks with default parameters (promoter regions were defined from −1 kb to +100 bp). We then combined the two annotated gene lists together. Motif finding was carried out with HOMER using the de novo setting for underpeaks area and known motif setting for promoter motif analysis. TFmotifView program (http://bardet.u-strasbg.fr/tfmotifview/) was used to verify the RelA 1 h and Rel 18 h peaks (Supplementary Fig. 1h). The programs 'findMotifsGenome.pl' and 'findMotifs.pl' were used to identify transcription factor binding motifs within peaks or promoter regions (−400 to +100 bp from TSSs). All samples were normalized to 10 million reads for visualization.

## Analysis of ATAC-seq data

Quality control of sequenced reads was performed by FastQC (v0.11.8) (https://www.bioinformatics.babraham.ac.uk/projects/fastqc/), and adapter filtration was performed by Trimmomatic (v0.33) (ref. [59]), with the Nextera paired-end adapter annotation provided in the software. Paired ends were aligned to mm9 using Bowtie2 (v2.9.2) (ref. [58]) with parameter -X 2000, which specifies that paired reads with insertion up to 2,000 bp were allowed to align. Reads with mapping quality score < 10 were filtered out using Samtools (v1.9), and PCR duplicates were removed using Picard tools (v1.127; http://broadinstitute.github.io/picard/). Model based analysis of ChIP-Seq (MACS) 2 (v2.1.1) (ref. [60]) was used for removing duplicate reads and calling peaks with parameter options −shift 100 −extsize 200. Differentially accessible peaks after anti-IgM treatment at 1 h, 4 h or 18 h were analyzed by DiffBind (v3.8.4; http://bioconductor.org/packages/release/bioc/vignettes/DiffBind/inst/doc/DiffBind.pdf), with FDR ≤ 0.05 and fold change ≥ 1.5 to define induced peaks at different time point compared with 0 h. HOMER findMotifsGenome.pl was used to investigate the motif enrichment of induced peaks compared with 0 h. For genome tracks, bigwig files were created from bam files using BEDTools[61] and UCSC Genome Browser Utilities[62]. Genome tracks were explored using the WashU Epigenome Browser[63].

## Heatmap visualizations

Heatmaps displaying normalized read densities of ATAC-seq and ChIP–seq samples were generated with the computeMatrix and plotHeatmap modules of the deepTools package v3.4.1 (ref. [64]).

## Droplet-based single-cell sequencing

Using a Chromium Single Cell 3′GEM Kit v3 (10x Genomics, 1000094), Chromium Single Cell 3′ Gel Bead Kit v3 (10x Genomics, 1000093) and the Chromium Chip B Single Cell Kit (10x Genomics, 1000153), the anti-IgM Fab′2 activated splenic B cell suspensions in RPMI plus 2.0% FBS, at 1,000 cells per µl, were loaded onto a chromium single-cell controller (10x Genomics) to generate single-cell gel beads-in-emulsion (GEMs) according to the manufacturer's protocol. Briefly, approximately 10,000 cells per sample ($n$ = 2) were added to chip B to create GEMs. Cells were lysed, and the bead captured poly(A) RNA was barcoded during reverse transcription in Thermo Fisher Scientific Veriti 96-well thermal cycler at 53 °C for 45 min, followed by 85 °C for 5 min. cDNA was generated and amplified. Quality control and quantification of the cDNA were conducted using Agilent's High Sensitivity DNA Kit (5067-4626) in the 2100 Bioanalyzer. scRNA-seq libraries were constructed using a Chromium Single Cell 3′Library Kit v3 (10x Genomics, 1000095) and indexed with Chromium i7 Multiplex Kit (10x Genomics, 220103). The libraries were sequenced using an Illumina HiSeq2500 sequencer with a paired-end, single-indexing strategy consisting of 28 cycles for read 1 and 91 cycles for read 2. Multiple sequencing runs were performed to achieve greater sequencing depth for each barcode.

## scRNA-seq data processing

Demultiplexing of raw base call files into FASTQ files was completed with 10x Genomics Cell Ranger (v.3.0.2) mkfastq coupled with mouse reference version mm10. Cell Ranger count combine was used to aggregate multiple sequencing run reads to achieve increased sequencing depth. The results from Cell Ranger were processed in R v3.6.3 with Seurat v3.2.3 (ref. 65) using default parameters, unless otherwise specified. Quality control filtering was applied to each sample to eliminate downstream analysis of empty droplets, low-quality cells and potential doublets. Filtering excluded genes detected in less than ten cells and cells containing more than 10% mitochondrial genes. For the *RelA*$^{fl/fl}$ and *RelA* cKO datasets, the filtering also excluded cells expressing less than 400 or more than 5,000 transcripts and below 800 or above 18,000 counts. While, for 0-, 1- and 4-h datasets, cells express less than 300 or more than 4,000 transcripts and below 800 or above 15,000 counts. For the WT and *Rel*$^{-/-}$ 18-h dataset, 300 or 6,000 transcripts and 800 or 40,000 counts, respectively, were filtered out. Subsequently, each sample was normalized using the NormalizeData function. The top 2,000 highly variable genes, selected with the FindVariableFeatures function, were used for integrated analysis. Following scaling, principal component analysis was performed, and then 20 dimensions were used for the FindNeighbors and RunUMAP functions to generate UMAP, setting a resolution of 0.3 to group cells into clusters with distinct gene expression profiles.

## Statistical analysis

Statistical tests were performed on qRT–PCR, ChIP–qPCR and flow cytometry data using GraphPad Prism 9.2.

## Reporting summary

Further information on research design is available in the Nature Portfolio Reporting Summary linked to this article.

## Data availability

The datasets generated and analyzed in this publication have been deposited in NCBI's Gene Expression Omnibus and are accessible through GEO Superseries accession number GSE197035 (https://www.ncbi.nlm.nih.gov/geo/query/acc.cgi?acc=GSE197035). Source data for Fig. 1c are provided. ChIP–seq data are also available at https://wangftp.wustl.edu/~dli/nih-mingming/biowulf/ and may be visualized at WashU Epigenome Browser: https://epigenomegateway.wustl.edu/. TFmotifView results are available here: http://bardet.u-strasbg.fr/tfmotifview/?results=734fKxPAnCh8w1 (RelA 1 h); http://bardet.u-strasbg.fr/tfmotifview/?results=H1bHoIdGkHrKPy (Rel 18 h). Supplementary Data 1–5 and Source data are provided with this paper.

## Code availability

No custom code was used in these studies.

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

## Acknowledgements

We thank all the members of Sen Lab in Gene Regulation Section (GRS) for valuable discussions. We thank M. Meffert (Johns Hopkins School of Medicine), C. Bassing (University of Pennsylvania) and W. Khan (University of Miami) for critical comments on the manuscript. We also thank E. Lehrmann for processing and tracking our GEO submittal. H.J. and W.Z. are partially supported by NIH grant R01HG009518. The remaining authors are supported by the Intramural Research Programs of the National Institute on Aging (NIA) and the National Institute of Allergy and Infectious Diseases (to M.Z. and P.L.).

## Author contributions

M.Z. and R.S. designed research; M.K. generated mice; M.Z. performed ATAC-seq, ChIP–seq, ChIP–PCR, bulk RNA-seq, western blots and

qRT–PCR experiments; C.A.S. performed bulk and scRNA-seq; J.J. carried out RNA analyses; A.S. performed FACS sorting and Amnis analysis; P.C. performed ChIP–PCR and RT–PCR; J.F. sequenced libraries; S.N. initiated studies with Tat-Cre. M.Z., S.D., C.A.S., Y.Z., K.M.-M., K.G.B., W.Z., P.L. and H.J. analyzed data, and M.Z., C.A.S., A.S. and R.S. prepared the manuscript.

## Competing interests

The authors declare no conflicts of interest.

## Additional information

**Extended data** is available for this paper at https://doi.org/10.1038/s41590-023-01561-7.

**Correspondence and requests for materials** should be addressed to Ranjan Sen.

**Extended Data Table 1 | Inducible number of genes and RelA or Rel binding across differentially expressed genes in each *k*-means category. Each category was further segregated into two groups with >5 or >2 and <5-fold change. Numbers correspond to RelA or Rel ChIP–seq peaks at the indicated times that were annotated to genes in each group by HOMER**

| Categories | Fold Change | Genes | RelA 1h | Rel 1h | RelA 4h | Rel 18h |
|---|---|---|---|---|---|---|
| a | ≥ 5 | 74 | 33 | 17 | 17 | 24 |
| a | ≥ 2 and < 5 | 235 | 88 | 49 | 47 | 80 |
| b | ≥ 5 | 140 | 10 | 6 | 8 | 17 |
| b | ≥ 2 and < 5 | 953 | 139 | 42 | 58 | 153 |
| c | ≥ 5 | 371 | 41 | 13 | 19 | 66 |
| c | ≥ 2 and < 5 | 1747 | 250 | 60 | 89 | 311 |
| d | ≥ 5 | 205 | 18 | 7 | 8 | 24 |
| d | ≥ 2 and < 5 | 529 | 82 | 25 | 35 | 75 |
| e | ≥ 5 | 57 | 5 | 2 | 2 | 7 |
| e | ≥ 2 and < 5 | 435 | 130 | 37 | 50 | 125 |
| f | ≥ 5 | 638 | 134 | 47 | 54 | 111 |
| f | ≥ 2 and < 5 | 1146 | 370 | 148 | 162 | 350 |

**Extended Data Table 2 | Direct NF-κB targets from Fig. 3d, cluster I3**

| Genes | RelA clusters | Rel clusters | Reported/Unknown target |
|---|---|---|---|
| 4632428N05Rik | #N/A | 2 | Unknown |
| A930024E05Rik | 3 | #N/A | Unknown |
| Arl5b | 3 | 2 | Unknown |
| Atf3 | #N/A | 2 | Reported |
| Atf4 | #N/A | #N/A | Unknown |
| B4galt5 | #N/A | 2 | Reported |
| B630005N14Rik | 3 | 2 | Unknown |
| Bend3 | #N/A | #N/A | Unknown |
| Bhlhe40 | 3 | 2 | Unknown |
| Blk | 5 | 5 | Reported |
| Bri3 | #N/A | #N/A | Unknown |
| Ccl3 | #N/A | #N/A | Reported |
| Ccnl1 | #N/A | 2 | Reported |
| Cd274 | 3 | 2 | Reported |
| Cd69 | #N/A | 2 | Reported |
| Cd83 | #N/A | 2 | Reported |
| Cd86 | #N/A | #N/A | Reported |
| Chd7 | #N/A | #N/A | Reported |
| Clk1 | #N/A | 5 | Unknown |
| Crem | #N/A | 5 | Reported |
| Csrnp1 | #N/A | #N/A | Reported |
| Ddit3 | #N/A | 2 | Reported |
| Dtx1 | 5 | 2 | Reported |
| Dusp1 | #N/A | 2 | Reported |
| Dusp10 | #N/A | 5 | Reported |
| Dusp2 | #N/A | 2 | Reported |
| E330020D12Rik | #N/A | #N/A | Unknown |
| Egr2 | #N/A | 2 | Reported |
| Egr3 | #N/A | 2 | Unknown |
| Ell2 | #N/A | 5 | Unknown |
| Epha2 | #N/A | #N/A | Reported |
| Fam167a | #N/A | 2 | Unknown |
| Fam43a | #N/A | 2 | Unknown |
| Fam76a | #N/A | #N/A | Unknown |
| Flcn | #N/A | #N/A | Unknown |
| Fos | #N/A | 2 | Reported |
| Fosb | #N/A | 2 | Reported |
| Gadd45b | 3 | 2 | Reported |
| Gch1 | #N/A | 2 | Unknown |
| Gem | #N/A | 2 | Reported |
| Gfi1 | #N/A | 2 | Unknown |
| Gm6377 | 3 | #N/A | Unknown |
| Gm9025 | #N/A | 5 | Unknown |
| Gucd1 | #N/A | #N/A | Reported |
| H2-Eb2 | #N/A | 5 | Unknown |
| Hilpda | #N/A | #N/A | Unknown |
| Ier2 | #N/A | 5 | Reported |
| Ifrd1 | #N/A | 2 | Unknown |
| Il1r2 | #N/A | #N/A | Reported |
| Ildr1 | #N/A | 5 | Unknown |
| Ilf3 | #N/A | 2 | Reported |
| Ing3 | 3 | 2 | Unknown |
| Irf4 | #N/A | 2 | Reported |
| Irs2 | #N/A | 2 | Reported |
| Junb | #N/A | 5 | Reported |
| Kdm6b | #N/A | 2 | Reported |
| Klf10 | #N/A | 2 | Reported |
| Lipg | #N/A | 2 | Reported |
| Litaf | #N/A | 2 | Reported |
| Lmna | 5 | #N/A | Reported |
| Maff | #N/A | #N/A | Reported |
| Mafk | #N/A | #N/A | Reported |
| Mapk6 | #N/A | 2 | Reported |
| Myc | #N/A | 2 | Reported |
| Nfkbia | 3 | 2 | Reported |
| Nfkbid | #N/A | 2 | Reported |
| Nfkbiz | #N/A | 2 | Reported |
| Nr4a2 | #N/A | #N/A | Reported |
| Nr4a3 | 5 | 2 | Reported |
| Orai1 | #N/A | 2 | Reported |
| Pdcd1lg2 | 3 | #N/A | Unknown |
| Pim1 | 3 | 5 | Reported |
| Plaur | 3 | 2 | Unknown |
| Plek | #N/A | 2 | Reported |
| Plekho2 | 3 | #N/A | Unknown |
| Plk3 | 5 | 2 | Reported |
| Ppp1r11 | 3 | 2 | Unknown |
| Ppp1r15a | #N/A | 2 | Reported |
| Ppp1r16b | #N/A | 2 | Unknown |
| Ptger4 | #N/A | 2 | Reported |
| Ptpn22 | #N/A | #N/A | Unknown |
| Rab20 | #N/A | #N/A | Unknown |
| Rabgef1 | #N/A | 2 | Unknown |
| Rap1b | 3 | 2 | Unknown |
| Rasgef1b | #N/A | 2 | Reported |
| Rel | 3 | 2 | Reported |
| Rgs1 | 3 | 2 | Unknown |
| Ripk2 | 3 | 2 | Reported |
| Rrad | #N/A | #N/A | Unknown |
| S1pr3 | #N/A | 2 | Unknown |
| Samsn1 | 3 | 2 | Reported |
| Sde2 | #N/A | #N/A | Unknown |
| Serpinf1 | #N/A | #N/A | Unknown |
| Slc16a10 | 5 | #N/A | Unknown |
| Slc41a1 | 5 | #N/A | Unknown |
| Spag9 | 5 | 2 | Unknown |
| Srgn | #N/A | 5 | Reported |
| Stap1 | #N/A | 2 | Unknown |
| Tagap | #N/A | 2 | Reported |
| Tatdn2 | #N/A | 2 | Unknown |
| Tgif1 | 3 | 2 | Unknown |
| Tgif2 | #N/A | #N/A | Unknown |
| Tnfaip3 | 3 | 2 | Reported |
| Tnfrsf1b | #N/A | 5 | Reported |
| Tob2 | #N/A | 2 | Unknown |
| Tpst2 | 5 | 2 | Unknown |
| Trac | #N/A | 2 | Unknown |
| Traf1 | 5 | 2 | Reported |
| Traf4 | #N/A | #N/A | Reported |
| Trib2 | #N/A | 2 | Unknown |
| Twsg1 | 5 | #N/A | Unknown |
| Zbtb10 | #N/A | 2 | Reported |
| Zfp36 | #N/A | 2 | Reported |

#N/A = not applicable

**Extended Data Table 3 | Direct NF-κB targets from Fig. 3d, cluster I1**

| Genes | Rel clusters | RelA clusters | Reported/ Unknown target | Genes | Rel clusters | RelA clusters | Reported/ Unknown target |
|---|---|---|---|---|---|---|---|
| 9130401M01Rik | #N/A | #N/A | Unknown | Ncf1 | 3 | #N/A | Unknown |
| Abcc4 | 4 | #N/A | Unknown | Nfkbie | 1 | #N/A | Reported |
| Amigo2 | 1 | #N/A | Reported | Nme6 | 4 | #N/A | Unknown |
| Aurkaip1 | 3 | #N/A | Unknown | Nup62 | 4 | #N/A | Reported |
| Banf1 | 4 | #N/A | Unknown | P4ha1 | #N/A | #N/A | Unknown |
| Batf | 4 | #N/A | Reported | Pde6d | #N/A | #N/A | Unknown |
| BC035044 | 4 | #N/A | Unknown | Phb2 | 4 | 1 | Unknown |
| Bcl2l1 | 4 | 1 | Reported | Plxnd1 | 4 | #N/A | Unknown |
| Cacybp | 4 | #N/A | Unknown | Pnkp | 3 | #N/A | Unknown |
| Canx | 4 | #N/A | Reported | Ppp2r5d | #N/A | #N/A | Unknown |
| Casp4 | 4 | #N/A | Reported | Psma4 | 4 | #N/A | Unknown |
| Cbx1 | 4 | #N/A | Unknown | Psma6 | 4 | #N/A | Unknown |
| Ccdc124 | 3 | #N/A | Unknown | Psma7 | 4 | #N/A | Unknown |
| Ccdc50 | #N/A | #N/A | Unknown | Psmd1 | 4 | #N/A | Unknown |
| Cct8 | 4 | #N/A | Unknown | Psmd12 | 4 | #N/A | Unknown |
| Cdc37 | #N/A | #N/A | Unknown | Psme2 | 4 | #N/A | Reported |
| Cdk4 | 4 | #N/A | Unknown | Psmg1 | 4 | #N/A | Unknown |
| Coro2a | 3 | #N/A | Unknown | Ptprs | 3 | #N/A | Unknown |
| Cyc1 | 4 | #N/A | Unknown | Ranbp1 | 4 | #N/A | Unknown |
| Dars | 4 | #N/A | Unknown | Rasal1 | 4 | 1 | Unknown |
| Dnaja1 | 4 | #N/A | Unknown | Rilpl2 | 3 | #N/A | Reported |
| Dnajc15 | 4 | #N/A | Unknown | Scarb1 | #N/A | #N/A | Reported |
| Dnmt3a | #N/A | #N/A | Unknown | Sf3a3 | 4 | #N/A | Unknown |
| Eif2s3x | 4 | #N/A | Unknown | Sfxn2 | 4 | #N/A | Unknown |
| Eif5a | 4 | #N/A | Unknown | Slc25a39 | #N/A | #N/A | Unknown |
| Emg1 | #N/A | #N/A | Unknown | Slc39a10 | #N/A | 1 | Unknown |
| Eno1 | 4 | #N/A | Unknown | Slfn2 | 3 | #N/A | Reported |
| Fam129a | 4 | #N/A | Unknown | Smc2 | 3 | #N/A | Reported |
| Farsb | 4 | #N/A | Unknown | Snrpa | #N/A | #N/A | Unknown |
| Fas | #N/A | #N/A | Reported | Spcs3 | 3 | #N/A | Unknown |
| Glrx3 | 4 | #N/A | Unknown | Srsf1 | 4 | #N/A | Unknown |
| Gm13835 | #N/A | #N/A | Unknown | St13 | 4 | #N/A | Unknown |
| Grap | 3 | 6 | Unknown | Stat1 | 4 | #N/A | Reported |
| Hivep3 | 1 | 1 | Unknown | Stat5a | 4 | #N/A | Reported |
| Hmgn3 | #N/A | #N/A | Unknown | Suclg1 | 3 | #N/A | Unknown |
| Hsp90ab1 | 4 | #N/A | Unknown | Supt16 | 4 | #N/A | Unknown |
| Hsp90b1 | 4 | #N/A | Unknown | Swap70 | 4 | #N/A | Unknown |
| Hspa8 | 4 | #N/A | Unknown | Tada1 | 4 | 2 | Unknown |
| Hspd1 | 4 | #N/A | Unknown | Tcp1 | 4 | #N/A | Unknown |
| Hspe1 | 4 | #N/A | Unknown | Tial1 | 4 | #N/A | Unknown |
| Icam1 | 4 | #N/A | Reported | Timm13 | 4 | #N/A | Reported |
| Il4i1 | 4 | #N/A | Reported | Trim30a | 4 | #N/A | Unknown |
| Irg1 | 4 | #N/A | Reported | Ubtf | 4 | #N/A | Reported |
| Kcnk5 | #N/A | #N/A | Reported | Uqcc2 | 4 | #N/A | Unknown |
| Kpnb1 | 4 | #N/A | Unknown | Wee1 | 3 | 1 | Unknown |
| Ldha | 4 | 2 | Unknown | Xrcc1 | 4 | #N/A | Unknown |
| Map2k1 | 4 | #N/A | Unknown | Xxylt1 | #N/A | #N/A | Unknown |
| Mlx | 3 | #N/A | Unknown | Zbp1 | 4 | #N/A | Unknown |
| Mrps5 | 4 | #N/A | Unknown | Zfp207 | 4 | #N/A | Unknown |
| Mthfd1 | 4 | #N/A | Unknown | Zranb2 | #N/A | #N/A | Unknown |
| Nampt | 4 | 6 | Reported | #N/A = not applicable | | | |

**Extended Data Table 4 | Direct NF-κB targets from Fig. 3d, cluster I6**

| Genes | Rel clusters | RelA clusters | Reported/ Unknown target | Genes | Rel clusters | RelA clusters | Reported/ Unknown target | Genes | Rel clusters | RelA clusters | Reported/ Unknown target |
|---|---|---|---|---|---|---|---|---|---|---|---|
| 0610010K14Rik | #N/A | #N/A | Unknown | Hist1h2ah | #N/A | #N/A | Unknown | Psme1 | 3 | #N/A | Reported |
| 1110059E24Ri | 6 | #N/A | Unknown | Hist1h2be | 3 | #N/A | Unknown | Ptbp1 | 3 | #N/A | Unknown |
| 1110059E24Rik | #N/A | 6 | Unknown | Homer1 | 4 | #N/A | Reported | Ptpn6 | 3 | #N/A | Reported |
| 1810037I17Rik | 3 | #N/A | Unknown | Idh2 | #N/A | #N/A | Unknown | Ptprcap | 3 | #N/A | Unknown |
| Actg1 | 3 | #N/A | Reported | Ifi27l2a | 3 | #N/A | Unknown | Rab8b | 3 | #N/A | Unknown |
| Adcy7 | 3 | #N/A | Unknown | Incenp | 3 | #N/A | Unknown | Rac2 | 3 | #N/A | Unknown |
| Adgre1 | 3 | 6 | Unknown | Inpp4a | #N/A | #N/A | Unknown | Rad51c | #N/A | #N/A | Unknown |
| Adk | #N/A | #N/A | Unknown | Jak3 | 3 | #N/A | Reported | Rbl1 | 3 | #N/A | Reported |
| Aldh9a1 | 3 | #N/A | Unknown | Kcnn4 | 6 | #N/A | Reported | Reep5 | 6 | #N/A | Unknown |
| Anp32a | 3 | #N/A | Unknown | Kif22 | #N/A | #N/A | Unknown | Rgs10 | 3 | #N/A | Unknown |
| Arhgap11a | 3 | #N/A | Unknown | Kif23 | 3 | #N/A | Reported | Rgs3 | 3 | #N/A | Reported |
| Arid3a | 3 | #N/A | Reported | Klhl9 | 3 | #N/A | Unknown | Rpl3 | 3 | #N/A | Unknown |
| Atp5d | 3 | #N/A | Unknown | Kmo | 3 | #N/A | Unknown | Rps2 | 3 | #N/A | Unknown |
| Atp5l | 3 | #N/A | Unknown | Lasp1 | #N/A | #N/A | Unknown | Rps6ka1 | #N/A | #N/A | Unknown |
| Atp5o | 3 | #N/A | Unknown | Lck | 3 | #N/A | Unknown | Rpsa | 3 | #N/A | Unknown |
| Axl | 3 | #N/A | Unknown | Lmnb1 | 3 | #N/A | Unknown | Rtca | 3 | #N/A | Unknown |
| BC028528 | #N/A | #N/A | Unknown | Lsp1 | 3 | 6 | Reported | Rxra | #N/A | #N/A | Unknown |
| Bcl11a | 3 | #N/A | Unknown | Ly86 | 3 | 6 | Unknown | S100a10 | 3 | #N/A | Reported |
| Bin2 | 3 | #N/A | Reported | Mcm5 | 3 | #N/A | Unknown | Sash1 | #N/A | #N/A | Reported |
| Birc5 | | #N/A | Reported | Mef2b | #N/A | #N/A | Reported | Selplg | #N/A | #N/A | Reported |
| Bub1b | 3 | #N/A | Unknown | Melk | #N/A | #N/A | Unknown | Sgol1 | #N/A | #N/A | Unknown |
| Capg | 3 | #N/A | Unknown | Mis18bp1 | 3 | #N/A | Unknown | Sh3kbp1 | 3 | #N/A | Unknown |
| Cbx3 | 3 | #N/A | Unknown | Mki67 | 3 | #N/A | Unknown | Shank1 | 3 | #N/A | Unknown |
| Ccnd2 | 3 | #N/A | Reported | Mrpl18 | 3 | #N/A | Unknown | Shisa8 | 3 | #N/A | Unknown |
| Ccr1 | 3 | #N/A | Reported | Mrpl45 | 3 | #N/A | Unknown | Siah2 | 2 | #N/A | Unknown |
| Cd81 | 3 | #N/A | Reported | Msh6 | 3 | #N/A | Unknown | Siglecg | #N/A | #N/A | Unknown |
| Cenpe | 3 | #N/A | Unknown | Mtmr4 | 3 | #N/A | Unknown | Sirpa | 3 | 6 | Reported |
| Cnp | 6 | #N/A | Unknown | Napsa | 3 | #N/A | Unknown | Slc25a4 | 3 | #N/A | Unknown |
| Commd1 | 3 | #N/A | Unknown | Ncf4 | 3 | 6 | Unknown | Snx3 | 3 | #N/A | Unknown |
| Commd10 | 3 | #N/A | Unknown | Nckipsd | #N/A | #N/A | Unknown | St14 | #N/A | #N/A | Unknown |
| Ctsc | 3 | #N/A | Unknown | Ndufa10 | 3 | #N/A | Unknown | St8sia6 | 3 | #N/A | Unknown |
| Dhrs3 | #N/A | #N/A | Unknown | Ndufa4 | 3 | #N/A | Unknown | Stim2 | #N/A | #N/A | Unknown |
| Dhx34 | #N/A | #N/A | Unknown | Nek2 | #N/A | #N/A | Reported | Tagln2 | 3 | #N/A | Unknown |
| E2f3 | 3 | #N/A | Reported | Nek7 | 3 | 6 | Reported | Tapbp | 3 | #N/A | Reported |
| Ebi3 | 3 | #N/A | Reported | Nfam1 | 3 | 4 | Unknown | Tbc1d10c | 3 | #N/A | Unknown |
| Eif2a | #N/A | #N/A | Unknown | Nrp1 | #N/A | #N/A | Unknown | Tle4 | 3 | #N/A | Unknown |
| Etv6 | #N/A | #N/A | Reported | Pcna | 3 | #N/A | Reported | Tlr2 | #N/A | #N/A | Reported |
| Evi2a | #N/A | 3 | Unknown | Pdia3 | 3 | #N/A | Unknown | Tmem154 | #N/A | #N/A | Unknown |
| Evl | 3 | #N/A | Unknown | Pfn1 | 3 | #N/A | Unknown | Tmem168 | 3 | #N/A | Unknown |
| Ezh2 | 3 | #N/A | Unknown | Pik3r6 | #N/A | #N/A | Unknown | Tmem229b | 3 | #N/A | Unknown |
| Fam111a | 3 | #N/A | Unknown | Pkm | 3 | #N/A | Reported | Tnfaip8 | 3 | 6 | Reported |
| Fam3c | 3 | #N/A | Unknown | Pld4 | 6 | #N/A | Unknown | Tnfrsf13b | 3 | #N/A | Unknown |
| Fcmr | 3 | 6 | Unknown | Plekhg2 | #N/A | #N/A | Unknown | Trim8 | #N/A | #N/A | Unknown |
| Fen1 | 3 | #N/A | Unknown | Plxnc1 | 3 | #N/A | Unknown | Tuba1b | 3 | #N/A | Unknown |
| Fgr | #N/A | 4 | Reported | Pml | #N/A | #N/A | Reported | Tubb5 | 3 | #N/A | Unknown |
| Fignl1 | #N/A | #N/A | Unknown | Pold1 | 3 | #N/A | Unknown | Txn1 | 3 | #N/A | Unknown |
| Grhpr | #N/A | #N/A | Unknown | Ppia | 3 | #N/A | Unknown | Tyk2 | #N/A | 6 | Unknown |
| Grn | 3 | #N/A | Unknown | Ppp1ca | 3 | #N/A | Unknown | Uchl3 | 3 | #N/A | Unknown |
| Gypc | #N/A | 6 | Reported | Prcp | 3 | #N/A | Unknown | Vav1 | 3 | #N/A | Unknown |
| Hhex | #N/A | #N/A | Unknown | Psmb8 | 3 | #N/A | Unknown | Zfp217 | 3 | #N/A | Unknown |
| Hic1 | 3 | #N/A | Unknown | Psmb9 | 3 | #N/A | Reported | #N/A = not applicable | | | |

**Extended Data Table 5 | Direct NF-κB targets from Fig. 3d, cluster I2**

| Genes | RelA clusters | Rel clusters | Reported/ Unknown target | Genes | RelA clusters | Rel clusters | Reported/ Unknown target |
|---|---|---|---|---|---|---|---|
| Aatf | #N/A | #N/A | Unknown | Ms4a4c | #N/A | #N/A | Unknown |
| Adm | #N/A | #N/A | Reported | Mthfd1l | #N/A | 1 | Unknown |
| Ahr | 2 | 1 | Unknown | Mvd | #N/A | #N/A | Unknown |
| Alcam | #N/A | #N/A | Reported | Naa50 | #N/A | #N/A | Unknown |
| Apex1 | 2 | 1 | Unknown | Nbn | #N/A | #N/A | Unknown |
| Arhgap39 | 2 | #N/A | Unknown | Nfkb1 | 1 | #N/A | Reported |
| Cd72 | 2 | 1 | Unknown | Nop16 | 2 | 1 | Unknown |
| Chd4 | #N/A | 4 | Reported | Nop58 | 2 | #N/A | Unknown |
| D17Wsu92e | #N/A | #N/A | Unknown | Nsun2 | 2 | 1 | Unknown |
| Dnase2a | #N/A | 1 | Reported | Nudt9 | #N/A | 4 | Unknown |
| Eif2s2 | #N/A | #N/A | Unknown | Oasl1 | #N/A | #N/A | Reported |
| Eif4a1 | #N/A | #N/A | Unknown | Pbdc1 | #N/A | #N/A | Unknown |
| Fabp5 | #N/A | #N/A | Unknown | Phgdh | #N/A | #N/A | Reported |
| Fam213a | #N/A | 4 | Unknown | Psme3 | 2 | #N/A | Unknown |
| Fchsd2 | 2 | 1 | Reported | Pus1 | 2 | #N/A | Reported |
| Fubp1 | #N/A | #N/A | Unknown | Rcc2 | 2 | 1 | Reported |
| Fyn | 2 | 1 | Unknown | Rnf213 | #N/A | #N/A | Unknown |
| G3bp1 | 2 | #N/A | Unknown | Serbp1 | #N/A | 1 | Unknown |
| Gbp5 | #N/A | #N/A | Reported | Setd8 | 2 | #N/A | Reported |
| Gclc | 2 | 1 | Reported | Shmt1 | 2 | 1 | Unknown |
| Gfer | #N/A | #N/A | Unknown | Slc16a1 | 2 | #N/A | Reported |
| Gid4 | #N/A | #N/A | Unknown | Slc17a5 | #N/A | 1 | Reported |
| Gpr65 | 2 | #N/A | Unknown | Smad3 | 2 | 1 | Reported |
| Hspa5 | #N/A | 4 | Unknown | Srsf7 | 1 | 4 | Unknown |
| Hspa9 | #N/A | 1 | Unknown | Tardbp | 1 | 1 | Unknown |
| Hyou1 | #N/A | #N/A | Reported | Tor3a | #N/A | #N/A | Reported |
| Il2ra | #N/A | 1 | Reported | Txnl4a | 2 | #N/A | Unknown |
| Kdm2b | #N/A | 1 | Reported | Utp18 | 2 | 1 | Unknown |
| Ltv1 | #N/A | 1 | Reported | Vegfa | #N/A | #N/A | Reported |
| M6pr | #N/A | #N/A | Unknown | Vprbp | #N/A | 4 | Unknown |
| Mat2a | 2 | 1 | Reported | Zc3h12c | #N/A | 1 | Reported |
| Mid1 | #N/A | 2 | Unknown | #N/A = not applicable | | | |
| Mreg | #N/A | 1 | Unknown | | | | |

**Extended Data Table 6 | Direct transcription factor targets of NF-κB**

| Early | Reported/ Unknown target | Late | Reported/ Unknown target |
|---|---|---|---|
| Atf3 | Reported | 0610010K14Rik | Unknown |
| Atf4 | Unknown | Aebp2 | Unknown |
| Bhlhe40 | Unknown | Arid3a | Reported |
| Chd7 | Reported | Batf | Reported |
| Crem | Reported | Bcl11a | Unknown |
| Egr2 | Reported | Bcl6b | Unknown |
| Egr3 | Unknown | E2f3 | Reported |
| Fos | Reported | Erf | Unknown |
| Fosb | Reported | Etv6 | Reported |
| Gfi1 | Unknown | Grhl1 | Unknown |
| Irf4 | Reported | Hhex | Unknown |
| Junb | Reported | Hivep3 | Unknown |
| Maf | Unknown | Hmgn3 | Unknown |
| Maff | Reported | Maz | Unknown |
| Mafk | Reported | Mis18bp1 | Unknown |
| Myc | Reported | Mlx | Unknown |
| Nfe2l2 | Reported | Nab2 | Reported |
| Nr4a1 | Reported | Nfya | Unknown |
| Nr4a2 | Reported | Rb1 | Unknown |
| Nr4a3 | Reported | Rbl1 | Reported |
| Tgif1 | Unknown | Rxra | Unknown |
| Tgif2 | Unknown | Stat1 | Reported |
| | | Stat5a | Reported |
| | | Ubtf | Reported |
| | | Zfp207 | Unknown |
| | | Zfp217 | Unknown |
| | | Zfp219 | Unknown |
| | | Zfp395 | Unknown |
| | | Zfp710 | Unknown |

# Reporting Summary

## Statistics

For all statistical analyses, confirm that the following items are present in the figure legend, table legend, main text, or Methods section.

| n/a | Confirmed | |
|---|---|---|
| ☐ | ☒ | The exact sample size (*n*) for each experimental group/condition, given as a discrete number and unit of measurement |
| ☐ | ☒ | A statement on whether measurements were taken from distinct samples or whether the same sample was measured repeatedly |
| ☐ | ☒ | The statistical test(s) used AND whether they are one- or two-sided<br>*Only common tests should be described solely by name; describe more complex techniques in the Methods section.* |
| ☒ | ☐ | A description of all covariates tested |
| ☒ | ☐ | A description of any assumptions or corrections, such as tests of normality and adjustment for multiple comparisons |
| ☐ | ☒ | A full description of the statistical parameters including central tendency (e.g. means) or other basic estimates (e.g. regression coefficient) AND variation (e.g. standard deviation) or associated estimates of uncertainty (e.g. confidence intervals) |
| ☐ | ☒ | For null hypothesis testing, the test statistic (e.g. *F*, *t*, *r*) with confidence intervals, effect sizes, degrees of freedom and *P* value noted<br>*Give P values as exact values whenever suitable.* |
| ☒ | ☐ | For Bayesian analysis, information on the choice of priors and Markov chain Monte Carlo settings |
| ☒ | ☐ | For hierarchical and complex designs, identification of the appropriate level for tests and full reporting of outcomes |
| ☐ | ☒ | Estimates of effect sizes (e.g. Cohen's *d*, Pearson's *r*), indicating how they were calculated |

*Our web collection on statistics for biologists contains articles on many of the points above.*

## Software and code

Policy information about availability of computer code

| | |
|---|---|
| Data collection | Sequencing data was collected on the HiSeq 2000 (all except the FO or MZ libraries) or the NovaSeq 6000 (FO and MZ libraries). BCL files were demultiplexed and converted to FASTQ files using Illumina's bcl2fastq program (v2.20.0.422). RT-PCR data was collected on the Thermo ABI ViiA 7 Real-Time PCR System using QuantStudio 1.6.1; Image Cytometry was collected on the Amnis ® ImageStream ® X Mk II Imaging Flow Cytometer; Flow cytometry was collected on either the BD Aria Fusion or BD Symphony flow cytometer. Immunoblotting images were collected on the Syngene Imaging System. |
| Data analysis | The software used for data analysis of ChIP-Seq, ATAC-Seq, bulk RNA-Seq and scRNA-Seq, Image cytometry, flow cytometry and RT-PCR are described in methods. No custom code or algorithms were utilized. ChIP-Seq - Bowtie1 V1.0, Bowtie2 V2.9.2, deepTools V3.4.1, HOMER V4.11 for motif discovery - all with default parameters; ATAC-Seq - FastQC V0.11.8, Trimmomatic V0.33 and Bowtie2 V2.9.2; bulk RNA-Seq - RSEM V1.3.2 , EBSeq V1.24.0, DESeq2 V1.22.0, ShinyGO 0.77 for pathway analysis, MATLAB V9.4 (R2018) for k-means clustering and DiffBind V3.8.4 - all with default parameters; scRNA-Seq - 10X Genomics Cell Ranger suite V.3.0.2 mkfastq, count and combine with standard parameters. Cell Ranger data were processed in R v3.6.3 with Seurat v3.2.3 using default parameters. Flow cytometry was analyzed using FlowJo v10.1 with default parameters. Image Cytometry was analysed using IDEAS 6.0 software (Luminex). GraphPad Prism V9.0 for graphs. |

For manuscripts utilizing custom algorithms or software that are central to the research but not yet described in published literature, software must be made available to editors and reviewers. We strongly encourage code deposition in a community repository (e.g. GitHub). See the Nature Portfolio guidelines for submitting code & software for further information.

## Data

Policy information about availability of data

All manuscripts must include a data availability statement. This statement should provide the following information, where applicable:
- Accession codes, unique identifiers, or web links for publicly available datasets
- A description of any restrictions on data availability
- For clinical datasets or third party data, please ensure that the statement adheres to our policy

Sequencing datasets generated and analyzed in this publication have been deposited in NCBI's Gene Expression Omnibus (Edgar et al., 2002) and are accessible through GEO Superseries accession number GSE197035 (https://www.ncbi.nlm.nih.gov/geo/query/acc.cgi?acc=GSE197035).  ChIP-Seq data is also available at https://wangftp.wustl.edu/~dli/nih-mingming/biowulf/ and may be visualized at WashU Epigenome Browser: https://epigenomegateway.wustl.edu/. TFmotifView results are available here: http://bardet.u-strasbg.fr/tfmotifview/?results=734fKxPAnCh8w1 (RelA 1h); http://bardet.u-strasbg.fr/tfmotifview/?results=H1bHoIdGkHrKPy (Rel 18h).

# Field-specific reporting

Please select the one below that is the best fit for your research. If you are not sure, read the appropriate sections before making your selection.

☒ Life sciences       ☐ Behavioural & social sciences       ☐ Ecological, evolutionary & environmental sciences

For a reference copy of the document with all sections, see nature.com/documents/nr-reporting-summary-flat.pdf

# Life sciences study design

All studies must disclose on these points even when the disclosure is negative.

| | |
|---|---|
| Sample size | ATAC-Seq, ChIP-Seq, bulk RNA-Seq, scRNA-Seq were all carried out with two biological replicates. No statistical methods were used to pre-determine sample sizes but our sample sizes are similar to those reported in previous publications (ATAC-Seq - doi.org/10.1016/j.immuni.2023.03.017; ChIP-Seq - DOI: 10.1016/j.immuni.2018.04.024; bulk RNA-Seq - DOI: 10.1126/science.abn7625; scRNA-Seq - DOI: 10.1016/j.immuni.2021.08.017 ) |
| Data exclusions | No data was excluded. |
| Replication | All the genomic assays were carried out with 2 biological replicates. Pearson correlation coefficients (r) between biological replicates were calculated.  For RelA and Rel ChIP-Seq, we only considered peaks that were present in both biological replicates. Differential analysis of RNA-Seq was carried out in DESeq2 using statistics to establish the reproducibility of significant differences. Differential analysis of ATAC-seqcarried out in Diffbind V? using statistics to establish the reproducibility of significant differences was  RT-qPCR assays were carried out with at least two independent RNA preparations and graphs provided show data point spread. |
| Randomization | B cells for each replicate of ChIP-Seq, ATAC-Seq, RNA-Seq, sc RNA-Seq were purified from 5 age-matched mice (mixed male/female).  We reasoned that additional randomization was not required for these studies because the data were obtained ex vivo from pooled B cells obtained from multiple mice. |
| Blinding | Data collection and analysis were not performed blind to the conditions of the experiments. |

# Reporting for specific materials, systems and methods

We require information from authors about some types of materials, experimental systems and methods used in many studies. Here, indicate whether each material, system or method listed is relevant to your study. If you are not sure if a list item applies to your research, read the appropriate section before selecting a response.

## Materials & experimental systems

| n/a | Involved in the study |
|---|---|
| ☐ | ☒ Antibodies |
| ☒ | ☐ Eukaryotic cell lines |
| ☒ | ☐ Palaeontology and archaeology |
| ☐ | ☒ Animals and other organisms |
| ☒ | ☐ Human research participants |
| ☒ | ☐ Clinical data |
| ☒ | ☐ Dual use research of concern |

## Methods

| n/a | Involved in the study |
|---|---|
| ☐ | ☒ ChIP-seq |
| ☐ | ☒ Flow cytometry |
| ☒ | ☐ MRI-based neuroimaging |

# Antibodies

| | |
|---|---|
| Antibodies used | rabbit-anti mouse RelA (8242S, Cell Signaling Technology); anti-RelA (sc-372, Santa Cruz Biotechnology); anti-Rel (sc-71, Santa Cruz Biotechnology); anti-hnRNP A1 (sc-32301, Santa Cruz Biotechnology); anti-H3K27ac (39133, Active Motif); anti-H3K4me1 (ab8895, Abcam); anti-H3K4me3 (ab8580, Abcam);Horseradish peroxidase coupled goat anti-mouse IgG (sc-2005, Santa Cruz Biotechnology), goat anti-rabbit IgG (sc-2004, Santa Cruz Biotechnology) and goat anti-mouse IgM F(ab`)2 (115-006-075, The Jackson Laboratory); B220+ (103240, BioLegend), AA4.1- (136510, BioLegend), CD21lo (123410, BioLegend) CD23hi(101606, BioLegend); Myc-AF-647 (1:50) (13871, Cell Signaling), Egr2-APC (1:50) (17-6691-82, eBiosciences), Rel-PE (1:200) (12-6111-80, eBiosciences), Ikba-PE (1:100) (7523, Cell Signaling), Dec1-AF-594 (1:100) (1800AF594, Novus Biologicals), Ezh2-PE (1:50) (562478, BD Bioscience), CD72-BV421 (1:200) (740058, BD Bioscience) and HSP90beta (rabbit polyclonal, Invitrogen); anti-rabbit Fab`2-PE antibody (1:1000) (79408, Cell Signaling Technology) |
| Validation | anti-H3K4me1 (ab8895, Abcam) is specific for mono-methylated Lysine 4 of histone H3 and does not recognize di- or tri-methyl Lysine 4, nor methylation at Lysine 9 per the manufacturer; anti-hnRNP A1 (sc-32301, Santa Cruz Biotechnology) as been utilized in 134 publications; anti-H3K27ac (39133, Active Motif)has been utilized in 77 publications; anti-H3K4me3 (ab8580, Abcam) has been utilized in 1741 publications; goat anti-mouse IgG (sc-2005, Santa Cruz Biotechnology) has been utilized in 929 publications and goat anti-rabbit IgG (sc-2004, Santa Cruz Biotechnology)has been utilized in 347 publications. anti-RelA (sc-372, Santa Cruz Biotechnology) was validated by the manufactor by western blot. anti-Rel (sc-71, Santa Cruz Biotechnology) was validated in our lab by western blot using Rel KO lysates (see source data). The manufacturer of the goat anti-mouse IgM F(ab`)2 (115-006-075, The Jackson Laboratory) antibody has been shown to react with the heavy chain of mouse IgM by ELISA. B220 (103240, BioLegend), RRID AB_11203896; AA4.1 (136510, BioLegend), RRID AB_2275868; CD21 (123410, BioLegend), RRID AB_940413; CD23(101606, BioLegend), RRID AB_312831; Myc-AF-647 (13871, Cell Signaling) has been utilized in 5 publications; Egr2-APC (17-6691-82, eBiosciences), RRID AB_11151502; Rel-PE (12-6111-80, eBiosciences), RRID AB_11042978; Ikba-PE (7523, Cell Signaling) has been utilized in 4 publications; Dec1-AF-594 (1800AF594, Novus Biologicals), RRID:AB_10000524; Ezh2-PE (562478, BD Bioscience), RRID AB_11152951; CD72-BV421 (740058, BD Bioscience), RRID AB_2739823; HSP90beta (PA3-012 rabbit polyclonal, Invitrogen); RRID:AB_2121220; anti-rabbit Fab`2-PE antibody (79408, Cell Signaling Technology) has been utilized in 4 publications. |

# Animals and other organisms

Policy information about studies involving animals; ARRIVE guidelines recommended for reporting animal research

| | |
|---|---|
| Laboratory animals | Species - Mus musculus Strains - WT (C57BL/6), RelA cKO (RelAfl/fl x Cd19-cre), RelAfl/fl  (Steinbrecher, Harmel-Laws et al. 2008), Cd19-cre (006785, The Jackson Laboratory) and Rel KO (Rel-/-)(Cheng, S., Hsia C.Y. et al. 2003); Age - 8-12 weeks; Sex - both sexes were used for experiments. |
| Wild animals | No wild animals were used. |
| Field-collected samples | No field-collected samples were used. |
| Ethics oversight | Institutional Animal Welfare Assurance Number – NIH Intramural Research Program – D16-00602; NIA AAALAC Unit Number – 000401. |

Note that full information on the approval of the study protocol must also be provided in the manuscript.

# ChIP-seq

## Data deposition

☒ Confirm that both raw and final processed data have been deposited in a public database such as GEO.

☒ Confirm that you have deposited or provided access to graph files (e.g. BED files) for the called peaks.

| | |
|---|---|
| Data access links<br>*May remain private before publication.* | GEO accession number specific to ChIP-Seq is GSE197032. |
| Files in database submission | RelA 0h1.fastq.gz, RelA 0h2.fastq.gz, RelA 1h1.fastq.gz, RelA 1h2.fastq.gz, RelA 4h1.fastq.gz, RelA 4h2.fastq.gz, Rel 0h1.fastq.gz, Rel 0h2.fastq.gz, Rel 1h1.fastq.gz, Rel 1h2.fastq.gz, Rel 18h1.fastq.gz, Rel 18h2.fastq.gz, H3K27ac 0h1.fastq.gz, H3K27ac 0h2.fastq.gz, H3K4me1 0h1.fastq.gz, H3K4me1 0h2.fastq.gz, H3K4me3 0h1.fastq.gz, H3K4me3 0h2.fastq.gz, H3K4me3 1h1.fastq.gz, H3K4me3 1h2.fastq.gz, H3K4me3 18h1.fastq.gz,H3K4me3 18h2.fastq.gz, RelA 0h1.bw, RelA 0h2.bw, RelA 1h1.bw, RelA 1h2.bw, RelA 4h1.bw, RelA 4h2.bw, Rel 0h1.bw, Rel 0h2.bw, Rel 1h1.bw, Rel 1h2.bw, Rel 18h1.bw, Rel 18h2.bw, H3K27ac 0h1.bw, H3K27ac 0h2.bw, H3K4me1 0h1.bw, H3K4me1 0h2.bw, H3K4me3 0h1.bw, H3K4me3 0h2.bw, H3K4me3 1h1.bw, H3K4me3 1h2.bw, H3K4me3 18h1.bw, H3K4me3 18h2.bw |
| Genome browser session<br>(e.g. UCSC) | WashU Epigenome Browser: https://epigenomegateway.wustl.edu/ ; data: https://wangftp.wustl.edu/~dli/nih-mingming/biowulf/ |

## Methodology

| | |
|---|---|
| Replicates | 2 replicates for each ChIP-seq sample |
| Sequencing depth | RelA ChIP-seq reads range: 18 million - 27 million; Rel ChIP-seq reads range: 10 million - 32 million; H3K27ac ChIP-seq reads range: 36 |

| | |
|---|---|
| Sequencing depth | million - 38 million; H3K4me3 ChIP-seq reads range: 17 million - 29 million; H3K4me1 ChIP-seq reads range: 36 million - 44 million. Total reads information for all samples are shown in Figure S1d and Figure S2a. |
| Antibodies | anti-RelA (sc-372, Santa Cruz Biotechnology); anti-Rel (sc-71, Santa Cruz Biotechnology); anti-H3K27ac (39133, Active Motif); anti-H3K4me1 (ab8895, Abcam); anti-H3K4me3 (ab8580, Abcam) |
| Peak calling parameters | CisGenome software (http://www.biostat.jhsph.edu/~hji/cisgenome/) was used for peak calling with a window size of 200 bases and a threshold of 4 reads (2 from each strand), FDR<0.1 and fold change >2. Peaks separated by less than 200 bases were merged into one peak. |
| Data quality | Quality check for both RelA and Rel ChIP-seq, was performed by 1) calculating Pearson correlation coefficient (r) between replicates and 2) comparing peak call numbers. RelA 0h replicates r=0.7; RelA1h replicates r=0.81; RelA 4h replicates r=0.77; Rel 0h replicates r=0.92;Rel 1h replicates r=0.93; Rel 18h replicates r=0.97; RelA 0h1 peak number=12484;RelA 0h2 peak number=5116; RelA 1h1 peak number=57105;RelA 1h2 peak number=39934; RelA 4h1 peak number=14461;RelA 4h2 peak number=21015; Rel 0h1 peak number=1353;Rel 0h2 peak number=780;Rel 1h1 peak number=2827;Rel 1h2 peak number=2548; Rel 18h1 peak number=15163; Rel 18h2 peak number=24410. For all other histone ChIP-seq we only did replicates pearson correlation, peak calling are not done on histone ChIP-seq, instead we use deepTools to see the heapmaps results of some specific regions. H3K4me3 0h replicates Pearson=0.98;H3K4me3 1h replicates Pearson=0.95;H3K4me3 18h replicates Pearson=0.99;H3K27ac 0h replicates Pearson=0.97; H3K4me1 0h replicates Pearson=0.93. |
| Software | Bowtie1 V1.0 software was used to map quality-filtered reads from demultiplexed FASTQ files to mouse genome assembly mm9 (Ensembl v67) with default options. CisGenome software (http://www.biostat.jhsph.edu/~hji/cisgenome/) was used for peak calling with a window size of 200 bases and a threshold of 4 reads (2 from each strand), FDR<0.1 and fold change >2. HOMER annotatePeaks.pl was used to annotate peaks with default parameters (promoter regions were defined from −1 kb to +100 bp). The programs findMotifs-Genome.pl and findMotifs.pl were used to identify transcription factor binding motifs within peaks or promoter regions . The computeMatrix and plotHeatmap modules of the deepTools package were used to produce heat map. |

## Flow Cytometry

### Plots

Confirm that:

☒ The axis labels state the marker and fluorochrome used (e.g. CD4-FITC).

☒ The axis scales are clearly visible. Include numbers along axes only for bottom left plot of group (a 'group' is an analysis of identical markers).

☒ All plots are contour plots with outliers or pseudocolor plots.

☒ A numerical value for number of cells or percentage (with statistics) is provided.

### Methodology

| | |
|---|---|
| Sample preparation | Isolated B cells (0.5x106) from spleens were fixed with 2% paraformaldehyde (15713, Electron Microscopy) for 10 mins in the dark at room temperature. Fixed cells were washed with PBS and resuspended in ice cold Perm Buffer III (558050, BD Biosciences) and incubated overnight at -20°C. Cells were washed with PBS and resuspended in BD perm/wash buffer (554723, BD Biosciences). |
| Instrument | BD FACS-Aria Fusion; BD Symphony |
| Software | FlowJo Version 10.1 |
| Cell population abundance | Splenic naïve B cells were comprised of 74.1 % follicular (FO) and 9.04 % marginal (MZ) cells |
| Gating strategy | Gating shown in the Supplementary figures S3c, S3g, and S4e |

☒ Tick this box to confirm that a figure exemplifying the gating strategy is provided in the Supplementary Information.

