## [Peer Review File · Nature Immunology]

Peer Review Information

Journal: Nature Immunology

Manuscript Title: NF- κ B subunits direct kinetically distinct transcriptional cascades in antigen receptor-activated B cells

Corresponding author name(s): Dr Ranjan Sen

Reviewer Comments & Decisions:

Decision Letter, initial version:
--

30th Mar 2022

Dear Ranjan,

Thank you for providing a point-by-point response to the referees comments on your manuscript entitled "NF- κ B subunits direct kinetically distinct transcriptional cascades in antigen receptor-activated B cells". As noted previously, that while they find your work of considerable potential interest, they have raised a number of concerns, some of which can be addressed by further analysis of datasets already in hand or further clarifications as indicated in your response to the referees. Please note, referee 1 was favorable for this study to be considered as a Resource for the immunology community. Much of the response to referee 3 involves clarifications, which we view as appropriate. Additional experimentation/analysis would be required to address some of the concerns articulated by referees 1 and 2. In light of these comments, we cannot accept the CURRENT manuscript for publication, but would be very interested in considering a revised version that addresses these concerns along the lines proposed in your rebuttal.

We invite you to submit a substantially revised manuscript, however please bear in mind that we will be reluctant to approach the referees again in the absence of major revisions.

Specifically, the revision should include new experiments/analysis to address:

- (1) Conditional deletion of RelA in B cells. I agree with your suggestion to use the Tam-inducible model of Cd20-TAM-Cre to delete RelA in the developing B cells, as this acute deletion strategy would circumvent compensation mechanisms that might confound subsequent analysis of the data. Additionally, it appears from the ImmGen database that Cd20 expression is not expressed in pro- or pre-B cells, but significantly increases after the pre-BCR checkpoint (aka Fraction E according to the Hardy scheme).
- (2) Further analysis of datasets in hand to identify potential pathways/programs affected by RelA & c-Rel activity during the early activation response in B cells.

(3) Validate changes in selected protein expression predicted by the transcriptomic analysis by flow cytometry and/or ImageStream analysis.

No need to perform additional experiments to address more mechanistic questions of how c-Rel might be suppressing RelA-dependent transcriptional targets (referee 2, points 3,4) as we agree that such experiments would be the nexus for more in-depth studies. Likewise, we waive additional ChIP-seq experiments for RelA at the 18h time point (referee 2, point 7), as little nuclear RelA is present at this later time point.

Please include the additional textual clarifications as indicated in your response letter.

When you revise your manuscript, please take into account all reviewer and editor comments, please highlight all changes in the manuscript text file in Microsoft Word format. Additionally, I would suggest changing the manuscript type to a Resource article (and emphasize in the manuscript text the Resource value of the study).

* If you have not done so already please begin to revise your manuscript so that it conforms to our Article format instructions at <http://www.nature.com/ni/authors/index.html>. Refer also to any guidelines provided in this letter.

The Reporting Summary can be found here:

[REDACTED]

If you wish to submit a suitably revised manuscript we would hope to receive it within 6 months. If you cannot send it within this time, please let us know. We will be happy to consider your revision so long as nothing similar has been accepted for publication at Nature Immunology or published elsewhere.

Nature Immunology is committed to improving transparency in authorship. As part of our efforts in this direction, we are now requesting that all authors identified as 'corresponding author' on published papers create and link their Open Researcher and Contributor Identifier (ORCID) with their account on the Manuscript Tracking System (MTS), prior to acceptance. ORCID helps the scientific community achieve unambiguous attribution of all scholarly contributions. You can create and link your ORCID from the home page of the MTS by clicking on 'Modify my Springer Nature account'. For more information please visit www.springernature.com/orcid.

Thank you for the opportunity to review your work.

Kind regards & safe travels,

Laurie

Laurie A. Dempsey, Ph.D.
Senior Editor
Nature Immunology
l.dempsey@us.nature.com
ORCID: 0000-0002-3304-796X

Referee expertise:

Referee #1: B cell biology

Referee #2: transcriptional regulation

Referee #3: transcriptional regulation

Reviewers' Comments:

Reviewer #1:

Remarks to the Author:

Comments to the authors;

Based on the previous nice series of Sen's group studies, here authors, by using sophisticated newly established technologies, have tried to understand in more details how orchestral NFkB activation takes place in BCR activated B cells. I think that the quality of the data is pretty high and that particularly scRNA seq analysis can clarify the previous unresolved cellular population issue, which is very nice. However, unfortunately, many of the conclusions are previously demonstrated (for instance RelA and Rel at early and late activation), although more solid conclusions are drawn by this study. One novel conclusion is functional antagonism of Rel and RelA. But this conclusion is only demonstrated by analysis of Rel ko B cells. More underlying mechanisms should be clarified. Moreover, because Rel is complete ko and RelA ko is made by CD19Cre, this might accumulate several gene transcription during B cell development. Authors should use CD23Cre or inducible Cre to delete Rel and RelA.

Reviewer #2:

Remarks to the Author:

Review of Zhao et al.

In their manuscript "NF-kappaB subunits direct kinetically distinct transcriptional cascades in antigen receptor-activated B cells", Zhao et al used ChIPseq, ATACseq and RNA sequencing to characterize RelA and c-Rel controlled gene expression programs in early B cell activation in response to engagement of the B cell receptor.

Their manuscript contains a large set of very valuable datasets that will be of great use for research in this field. The authors use an IKK2 inhibitor as well as RelA and c-Rel knockout B cells to assess the respective effects on gene expression by RNA sequencing. Through their comparative analyses they identify a subset of RelA regulated genes whose expression increases in the absence of c-Rel, indicating that the latter acts to downmodulate these genes at early time-points (and vice versa to a much lesser extent). They also suggest that RelA has a higher transactivation potential for a subgroup of genes, including Myc and Irf4. They identify early and late NF-kappaB target genes and suggest a marked heterogeneity with respect to their expression at the single cell level. The experiments are comprehensive and overall very well done.

However, I have the impression that inspite of the wealth of data the conclusions drawn are very technical. Many different clusters and subclusters are defined and correlated with each other. I miss attempts to integrate these data into the biology of B cell activation and function. Zhao et al suggest that their data imply a mutual antagonism of RelA and c-Rel at early time-points of B cells activation, with c-Rel being the main suppressor. But no attempt is being made to verify this notion or to explore the potential functions. Furthermore, there is no exploration of what the temporally sequential subdivision of NF-kappaB subunit responses could mean for physiological B cell responses (in living organisms) or whether the extracted gene patterns/sets could be used to identify stages of B cell activation in vivo. The connection to malignancy appears somewhat superficial to me. The single cell data appear to me underanalyzed. Surely more information can be extracted from these datasets. Overall, general key conclusions regarding NF-kappaB function or the regulation of B cell activation resulting from this work did not become clear to me.

I have the following major comments and suggestions:

- 1) In general, a biological interpretation of the data should be strengthened. What gene programs are up and down-regulated at which time-points controlling what cellular/biological responses?
- 2) What is the biological meaning of the temporal subdivision of labor between RelA and c-Rel? What gene functional expression programs are guided through this sequence of events and why?
- 3) How does c-Rel antagonize RelA-dependent gene expression? Functional experiments addressing this issue would strengthen the manuscript. This could be mechanistically investigated at the example of *Nfkbia*, amongst other genes.
- 4) The authors describe a c-Rel dominated gene expression including mostly novel NF-kappaB-dependent genes at later time-points. This is an interesting finding that would warrant follow up interrogation: is the gene expression program c-Rel dominated due to the loss of RelA activation? If RelA could be active in the nucleus (for example through overexpression of a variant lacking the nuclear export signal) would it also bind these genes and activate a similar gene expression program (maybe to be investigated in c-Rel knockout B cells)? Or is it due to a biochemical difference between RelA and c-Rel (differences in transcriptional activation? How strong is this effect? Do co-factors play a role that are not present at early time-points or do not efficiently bind RelA as suggested by the authors in the discussion? The authors implicate *Irf* genes, which could be tested for such a role by gain and loss of function experiments.
- 5) I lack some additional information/validation in order to judge the interpretation of the single cell RNAseq analyses. How homogenous is the activated B cell population isolated *ex vivo*? Could differences between marginal zone B cells, transitional B cell subsets etc contribute to the observed effects? Some of the statements of cellular expression of genes could be caused by mRNA detection levels in these systems rather than reflect cellular on/off situations. Key statements have to be validated by other techniques, for example intracellular flow cytometry for proteins (*BclxL*, *Myc*) and mRNA.
- 6) A characterization of BCR activation and the NF-kappaB response at the single cell is missing. We do not know how homogeneously the B cells are activated in the system and how (homogeneously) BCR activation translates into NF-kappaB activation at a single cell level. However, this information would in my eyes be crucial to correlate B cell activation (homogenous or not) to RNAseq and scRNAseq results at different time-points. The authors should measure RelA and c-Rel expression and nuclear localization at the single cell level during the time course together with other general parameters of cellular activation. Are there strong differences in cell to cell activation at different time points to reflect different activation states at the transcriptional level, as suggested by the scRNAseq? If not, where does the transcriptional heterogeneity come from?
- 7) In the ChIPseq analyses a direct comparison between RelA and c-Rel bound genes is only possible at the 1h time-point. That limits the interpretation of their respective direct targets during the early activation phase (and also the comparison to the RNAseq data at the different time-points). It would be more informative to have the ChIPseq for both factors during all time-points. This also applies to the validation experiments shown in Figure 1d.

Minor concerns and suggestions:

8) Can the early and late gene expression signatures be identified in early in vivo B cell responses?

9) In Figure 1d it seems that an average of two experiments are shown, together with the standard error of the mean? I do not think that SEM is an adequate descriptor in this case. Both (all) data points should be shown instead and a higher number of replicates should be produced. Furthermore, more Rel and RelA target genes should be investigated here to give an impression of the data.

10) There are ample examples of c-Myc regulation especially by c-Rel in the literature. Therefore it is somewhat surprising to have it classified as a Rel-repressed gene here. This should be validated by other means, as the scRNAseq does not seem to show major (meaningful?) differences. What does the transient induction from 1h to 4h of Myc mean functionally? How does this relate to increased late gene expression specifically through c-Rel, which repressed RelA-mediated Myc activation, what are the biological effects of the Myc-regulated genes?

11) A Comparison of present datasets with other publicly available datasets on B cell transcriptomes influenced by loss or gain of RelA or c-Rel is missing.

12) How many cells are depicted in Figure S5a? Why are there such differences between RelA/F and WT samples? If these are due to batch effects, can they be corrected to allow an integrative analysis of the whole dataset?

13) As noted by the authors, the incomplete inactivation of RelA in CD19Cre RelA/F B cells affects their dataset. The authors could consider employing Mb1Cre instead to obtain much higher RelA knockout proportions.

Reviewer #3:

Remarks to the Author:

The manuscript by Zhao et al. addresses the important problem of distinguishing selectivity in NF- κ B signaling by focusing on the binding and transcriptional response of RelA and Rel in stimulated B cells. They integrate a diverse set of 'omic' datasets to address direct targets of each subunit and their temporal roles in the B cell response. While I think the question is important, I found the manuscript quite descriptive and less focused on particular questions. The authors describe a diverse set of findings from motifs, to gene expression clusters, to impact of knockouts, but it was hard to follow the narrative at times. A more focused reworking of the manuscript would likely help the readability help the reader better focus in on the critical conclusions. However, more concerning is it is not clear in its current state how this paper moves the field beyond what was already known. Many ChIP-seq papers have been published on NF- κ B factors, and many studies of the temporal impact of NF- κ B, but there is little discussion of these results and therefore it is hard to assess the authors conclusion that this is the first demonstration of kinetically distinct cascades of gene expression in activated B cells. The conclusion about functional antagonism between RelA and Rel is also complicated by potential confounding effects (see detailed comments below). There are also a number of concerns about the motif analysis and the conclusions drawn from that in its current form.

Specific issues/concerns/comments:

Page 6: Are the numbers of RelA and Rel peaks in keeping with what others have found? While I understand that the cell types may matter but Zhao et al. (PMID: 25159142), for example, found considerably more sites in resting lymphoblastoid cells (e.g., 20,067 RelA, 6,765 Rel peaks). There a number of other NF- κ B ChIP-seq datasets, so it would be helpful for the authors to comment on the numbers of peaks given the sensitivity to protocol and antibodies. Also, was the irreducible discovery rate (IDR) method (ENCODE consortium best practices) used for the replicate averaging?

Page 6: Was motif finding performed on promoters only? It wasn't exactly clear from the methods.

Page 6: The limitation to the top 3 motifs seems artificial. Towards a more encompassing and transparent analysis, I encourage the authors to list all significant motifs identified in their ChIP-seq peaks. If this becomes significantly more than 3 for each experiment, some form of a heatmap could be used to show them. I would also like to see (data can be in Supplement) that an independent motif analysis program (e.g., Centrimo from the MEME suite) arrives at similar conclusions for the motif analysis.

Page 7: "We conclude that the first phase of NF- κ B induced signaling leads to widespread recruitment of RelA genome-wide and more restricted recruitment of Rel to mostly overlapping sites." Why is this 'restricted'? In other words, is this just because it's at lower concentration (i.e., it binds higher later). To help address this, it would be helpful if the authors could include some indication of the RelA and Rel protein (ideally nuclear) concentration along with the ChIP-seq peak data in Figure 1.

Page 7: The authors note - "The IRF motif (alone or combined with PU.1) was not included in the top three RelA-specific binding regions" First, the syntax is confusing, it should probably not be 'RelA-specific binding regions', but 'RelA-specific binding motifs'.

Page 7: "Promoter-associated RelA binding sites were were marked" - the word 'were' is repeated.

Page 7: "From time-dependent ATAC-Seq, we identified several thousand sites at which pre-existing chromatin accessibility was transiently increased ...We conclude that transiently bound RelA at these regions alters chromatin structure induced by constitutively expressed factors" While this seems reasonable, how do the authors determine when a difference in ATAC-seq peak is significant? Was a particular threshold for differential used? Further, how do the authors rule out the possibility that another TF is also binding to these peaks and leading to the observed changes?

Page 8: "The timing of gene expression changes mostly coincided with or followed changes in chromatin accessibility (Figure 3b)." Is this accessibility of the promoters? Please indicate in main text.

Page 8: "We observed inducible RelA binding across 39% of genes in category 'a' (rapidly and transiently induced) and 13% of genes in categories 'b' and 'c' (Figure 3d)." What is meant by 'binding across genes'? Is this the promoters of the genes? Across the gene body?

Page 9: "189 transiently induced RNAs were suppressed by the inhibitor (Figure 3e, cluster I3); 117 of these genes bound RelA at 1h, implicating them as direct NF- κ B targets (Table S1)." Again, does this mean the gene promoter was bound by RelA? Please re-word these as it is confusing what is meant by a gene binding to a TF. (There are a number of these instances throughout the paper, it would be

more clear if the authors could re-word).

Page 9: When is the BAY inhibitor given to the cells to identify the intermediate and late-phase activated genes? Is it just before that time point, or is it before induction? If it is before the IgM treatment, then it is not clear how the author rule out indirect effects of other late-phase TFs in the regulation of those target genes. For the 1h time-point it is similarly a potential issue, but perhaps less so.

Page 10: "Absence of Rel also increased inducible activation of many genes at 1h (Figure 3g, cluster R2), suggesting widespread repressive role for this family" This conclusion is complicated as there could be a number of compensatory mechanisms that result in the cells that have Rel knocked out. To confirm this conclusion it would be important to show that over-expression of cRel (ideally in an inducible manner) suppressed these genes.

Page 11: "To the best of our knowledge, this is the first demonstration of the differential activation potential of RelA or Rel on endogenous genes." As above, this interpretation is complicated by the possible compensatory mechanisms in the knockout cells. Over expressing each TF in an inducible manner – and showing that it leads to opposite effects on these genes – would seem to more directly address this point of activation potential. This is an intriguing idea, and one that appears to occur in a gene/loci-specific manner, so a more direct experiment would be very helpful to support it.

Author Rebuttal to Initial comments

February 21, 2023

Dr. Laurie A. Dempsey
Senior Editor
Nature Immunology
One New York Plaza Suite 4600
New York, NY 10004-1562
USA

Dear Laurie:

Thank you for coordinating the review of our manuscript 'NF- κ B subunits direct kinetically distinct transcriptional cascades in antigen receptor-activated B cells' by Zhao et al. (NI-A33546A). We were encouraged by the positive tone of Reviewer comments and sincerely appreciate the opportunity to revise the manuscript. I am now submitting such a revised version

to be considered for publication in *Nature Immunology*. Before highlighting the extensive changes made in response to your summary and Reviewer comments, I would like to apologize for going several months beyond the time I anticipated would be required to substantively address all comments. This was, in large part, due to complications that arose after we had spent considerable time generating *RelA^{fl/fl}xCd20-Tam-Cre* mice to provide an independent measure of the effects of deleting *RelA*. Consequently, we had to optimize an alternate method to accomplish the same objective. As shown in a new Figure S4a-b, use of Tat-Cre to delete *RelA* in mature B cells corroborated our earlier conclusions based on deleting *RelA* with *Cd19-cre*.

Beyond this critical experiment we made every effort to address all Reviewer comments in the revised manuscript. At the experimental level these include new ChIP studies to validate unique and shared sites of *RelA*- or *Rel* recruitment genome wide, analysis of protein expression of some NF- κ B target genes, determining the proportion of cells that induce NF- κ B translocation in BCR-activated B cells, new ChIP studies to validate *Rel* binding to late activated genes, and time-dependent transcriptional analyses of purified follicular and marginal zone B cells. At the analytic level, we include KEGG pathway analyses of early and late NF- κ B target genes in BCR-activated B cells, IDR analyses of ChIP-Seq replicates, evaluation of transcription factor motif enrichment by a different method than HOMER, and more complete lists of significantly enriched motifs associated with *RelA*- and *Rel* binding sites genome wide.

Point by point responses to editorial and reviewer comments follow (comments are underlined and our responses noted thereafter). All changes in the text and figure legends are noted in red font.

Editorial comments

- 1) 'Conditional deletion of *RelA* in B cells...(the goal was to use an) acute deletion strategy that would circumvent compensation mechanisms that might confound subsequent analysis of the data'

We first proposed to do this by generating *RelA^{fl/fl}xCd20-Tam-cre* mice in which *RelA* could be deleted by tamoxifen treatment of mature B cells. However, when we obtained the correct genotype after several months of breeding, we found that splenic B cells did not express cell surface Cd20 as expected based on the original transgene design. Additionally, *ex vivo* tamoxifen treatment did not efficiently delete the *RelA* gene. After some trouble shooting, we decided to put this method on hold and adopt an alternate strategy towards the same conceptual goal. For this we treated *RelA^{fl/fl}* (or control C57BL/6) B cells with endotoxin-free Tat-cre to induce gene deletion. After optimizing concentration and duration of treatment we obtained approximately 70-80% loss of *RelA* protein in B cells from *RelA^{fl/fl}* mice (new Figure S4a). we used these cells and identically treated C57BL/6 B cells for *in vitro* activation and quantitative RT-PCR analyses of *RelA*-selective NF- κ B target genes. As shown in new Figure S4b, the results closely paralleled those of RNA-Seq studies in *RelA^{fl/fl}xCd19-cre* B cells (discussed on page 12 of text). We conclude that early developmental deletion of *RelA* with *Cd19-cre* does not

induce compensatory mechanisms that affect RNA analyses in mature B cells. We surmise this is because Rel compensates effectively for loss of RelA during development.

2) ‘Further analysis of datasets in hand to identify potential pathways/programs affected by RelA and Rel activity during early activation response in B cells’.

In the revised manuscript we provide KEGG pathway analyses for early and late NF- κ B responsive genes in BCR-activated B cells (new Figure S3d). Pathway analyses were carried out in two steps. First, we examined genes that were downregulated in cells treated with the IKK2 inhibitor (that blocks early RelA/Rel and late Rel induction). Early/transiently induced genes enriched for pathways such as ‘NF- κ B signaling’, ‘TNF signaling’ and ‘MAPK signaling’ (new Figure S3d, top panel, cluster I3). By contrast, late NF- κ B responsive genes enriched for pathways associated with neurodegenerative diseases, metabolic pathways, ‘DNA replication’ and ‘Cell cycle’ (new Figure S3d, top panel, clusters I1 and I6). Second, we examined direct NF- κ B target genes in each category (new Figure S3d, bottom panel). We found that pathways associated with each category were similar to those noted with the larger NF- κ B responsive gene sets. These observations demonstrate that genes in early/transient (I3), continuously induced (I1) and late induced (I6) clusters have distinct biological functions (discussed on page 10 of text).

We also carried out KEGG pathway analyses with genes induced with different kinetic profiles in purified follicular and marginal zone B cells (new Figure S4h). A key distinction between these closely related B cell subsets was absence of ‘NF- κ B signaling’ and ‘TNF signaling’ pathways (amongst others) in early/transiently induced genes in marginal zone B cells. We verified that early NF- κ B signaling was attenuated in marginal zone B cells by quantitative RT-PCR assays of select NF- κ B target genes (new Figure S4i). These studies indicate that our multi-omics data reflect follicular B cells (discussed on page 13 of text).

3) ‘Validate changes in select protein expression predicted by transcriptomic analysis by flow cytometry and/or ImageStream analysis.’

Results of protein analysis by flow cytometry are shown in new Figure S3c and discussed on page 10 of the text. For these studies we examined 4 known NF- κ B target genes and 4 new targets predicted by our combination ChIP-Seq/RNA-Seq/NF- κ B inhibition studies. Note that protein data is scant even for known target genes since the vast majority of studies do not examine the effects of transcriptomic changes on protein expression. Despite protein expression analyses being limited by availability of flow cytometry compatible antibodies, we noted several interesting features. First, other than for Myc, peak protein expression of early activated genes (such as *Egr2*, *Rel* and *Nfkb1a*) occurred at late time points (new Figure S3c, left panel). Second, blocking NF- κ B activation with the IKK2 inhibitor reduced protein expression to varying degrees, presumably indicating additional post-transcriptional regulatory programs at play. Lack of I κ B α induction at the 4h time point likely reflects the combined effects of BCR-induced

degradation and new protein synthesis. Third, protein expression of late induced genes occurred at 18h and was reduced in IKK2 inhibitor treated cells. We hypothesize that the varying degrees of protein suppression observed reflect post transcriptional/translational regulatory mechanisms. For example, the small effect of NF- κ B inhibition on Hsp90b1 expression may be due to a) increased translation of pre-existing mRNA and/or b) elevation of *Hsp90b* mRNA by other (non-NF- κ B-dependent) pathways. As might be expected of a stress sensor, a lot of NF- κ B-dependent transcriptional regulation is super-posed on other pathways in activated cells. In other words, only a subset of NF- κ B target genes show an ‘all or none’ effect in RelA- or Rel-deficient cells, or in the absence of NF- κ B activation. The remaining genes, whose mRNA levels are modulated by NF- κ B, are nevertheless NF- κ B target genes.

Response to reviewers

Reviewer #1

We thank the reviewer for noting that ‘the quality of the data is pretty high and that particularly scRNA seq analysis can clarify the previous unresolved cellular population issue, which is very nice.’ However, he/she also raised some concerns as outlined below.

- 1) ‘However, unfortunately, many of the conclusions are previously demonstrated (for instance RelA and Rel at early and late activation), although more solid conclusions are drawn by this study.’

We agree that there have been sporadic examples of genes that are activated by late Rel. However, we believe that this temporal distinction between RelA and Rel has not been systematically studied. We did not find publications that the reviewer may have had in mind and recent reviews do not make this distinction accepted fact. Indeed, even this reviewer concludes that ‘more solid conclusions are drawn by this study’. We believe that our study will firmly place this concept in the framework of NF- κ B gene regulation studies.

- 2) ‘One novel conclusion is functional antagonism of Rel and RelA. But this conclusion is only demonstrated by analysis of Rel KO B cells. More underlying mechanisms should be clarified.’

Several interesting features emerged from our analyses, including functional antagonism between RelA and Rel when both are present in the nucleus. We previously quoted a paper that purports to find Rel associated with co-repressor complexes but did not follow up on this line of inquiry. Rather, as noted in the revised version we favor the idea that antagonism results from competition between RelA and Rel for the same binding sites, with RelA being a stronger transcriptional activator (page 15). We believe that mechanistic experiments regarding this and other interesting observations are not the purview of this manuscript.

- 3) ‘Because Rel is complete KO and RelA KO is made by CD19Cre, this might accumulate several gene transcription during B cell development. Authors should use CD23Cre or inducible Cre to delete Rel and RelA.’

We noted in the previous version of the manuscript that complete or conditional deletion of *Rel* has identical phenotypes. As described in the response to the editor, we encountered problems with the use of *Cd20-Tam-cre* to acutely delete RelA in mature B cells and thereby circumvent compensatory mechanisms that may arise from developmentally early deletion of RelA. Instead, in the revised manuscript we provide analysis of RelA-deficient B cells obtained by treating mature B cells from *RelA^{fl/fl}* mice with Tat-cre. The results closely corroborate studies with *RelA^{fl/fl}xCd19-cre* B cells (new Figure S4 a and b, discussed on page 12 of text).

Reviewer #2

This reviewer found our ‘manuscript contains a large set of very valuable datasets that will be of great use for research in this field’.... and that ‘the experiments are comprehensive and overall very well done.’ We appreciate the positive sentiments expressed here. However, the reviewer was concerned that we had not explored the implications of our findings more comprehensively. He/she had several comments and suggestions as outlined below.

Major comments

- 1) ‘In general, a biological interpretation of the data should be strengthened. What gene programs are up and down-regulated at which time-points controlling what cellular/biological responses?’

We now provide KEGG pathway analyses for early and late NF-κB target genes (new Figure S3d). These analyses demonstrate that genes in early/transient (I3), continuously induced (I1) and late induced (I6) clusters have distinct biological functions (discussed on page 10 of text). See also response #2 to Editor.

- 2) ‘What is the biological meaning of the temporal subdivision of labor between RelA and c-Rel? What gene functional expression programs are guided through this sequence of events and why?’

See response 1) above and response #2 to Editor.

- 3) ‘How does c-Rel antagonize RelA-dependent gene expression? Functional experiments addressing this issue would strengthen the manuscript. This could be mechanistically investigated at the example of Nfkb1a, amongst other genes.’

As outlined in response to Reviewer 1, the goal of the current studies was not to carry out mechanistic analyses on the several interesting conclusions that arose from the data. This is clearly an important point that will be addressed in future studies.

- 4) ‘The authors describe a c-Rel dominated gene expression including mostly novel NF-kappaB-dependent genes at later time-points. This is an interesting finding that would warrant follow up interrogation: is the gene expression program c-Rel dominated due to the loss of RelA activation? If RelA could be active in the nucleus (for example through overexpression of a variant lacking the nuclear export signal) would it also bind these genes and activate a similar gene expression program (maybe to be investigated in c-Rel knockout B cells)? Or is it due to a biochemical difference between RelA and c-Rel (differences in transcriptional activation? How strong is this effect?)? Do co-factors play a role that are not present at early time-points or do not efficiently bind RelA as suggested by the authors in the discussion? The authors implicate Irf genes, which could be tested for such a role by gain and loss of function experiments.’

The reviewer brings up many interesting ideas that are worthy of further exploration. As described in response to #3, however, such studies were felt to be beyond the purview of the current manuscript. The questions raised represent comprehensive analyses which, in our opinion, are best left for future studies.

- 5) ‘I lack some additional information/validation in order to judge the interpretation of the single cell RNAseq analyses. How homogenous is the activated B cell population isolated ex vivo? Could differences between marginal zone B cells, transitional B cell subsets etc contribute to the observed effects? Some of the statements of cellular expression of genes could be caused by mRNA detection levels in these systems rather than reflect cellular on/off situations. Key statements have to be validated by other techniques, for example intracellular flow cytometry for proteins (Bclxl, Myc) and mRNA.’

Our single cell assays were carried out to gain further insights into the results of bulk RNA-Seq from naïve spleen B cells. They demonstrated that trends observed in bulk RNA-Seq, including kinetics and RelA- or Rel dependency were recapitulated in single cell assays. Transcriptional antagonism between Rel and RelA was also clearly evident at the single cell level. We concluded that the observed trends reflected gene expression changes on a per cell basis.

To further address these points, we carried out RNA-Seq with purified follicular and marginal zone B cells. The resulting data indicate that our multi-omics assays largely reflect responses of follicular B cells. We also noted in both bulk and single cell RNA-Seq that marginal zone B cells had attenuated early NF-κB responses (new Figures S4e, f, g, h and i; discussed on page 13 of text).

Intracellular flow cytometry of several NF-κB target genes is included in the revised manuscript (new Figure S3c). See also response #3 to Editor.

- 6) ‘A characterization of BCR activation and the NF-kappaB response at the single cell is missing. We do not know how homogeneously the B cells are activated in the system and how (homogeneously) BCR activation translates into NF-kappaB activation at a single cell

level....The authors should measure RelA and c-Rel expression and nuclear localization at the single cell level during the time course together with other general parameters of cellular activation....If not, where does the transcriptional heterogeneity come from?'

To explore the heterogeneity of the B cell population to NF- κ B activation we carried out Amnis ImageStream analysis of RelA translocation in response to anti-IgM. Under optimal anti-IgM conditions, we found approximately 50% of cells undergo RelA nuclear translocation (discussed on page 6 of text).

- 7) 'In the ChIPseq analyses a direct comparison between RelA and c-Rel bound genes is only possible at the 1h time-point....It would be more informative to have the ChIPseq for both factors during all time-points. This also applies to the validation experiments shown in Figure 1d.'

We did not carry out RelA ChIP-Seq at 18h because there is very little nuclear RelA at this time point.

Minor concerns and suggestions:

- 8) 'Can the early and late gene expression signatures be identified in early in vivo B cell responses?'

Another interesting question, but one that we think goes beyond the focus of the current manuscript which is to rigorously identify targets of early and late NF- κ B activation in B cells.

- 9) 'In Figure 1d it seems that an average of two experiments are shown, together with the standard error of the mean? I do not think that SEM is an adequate descriptor in this case. Both (all) data points should be shown instead and a higher number of replicates should be produced. Furthermore, more Rel and RelA target genes should be investigated here to give an impression of the data.'

Additional examples are included in the revised manuscript (new Figure 1c).

- 10) 'There are ample examples of c-Myc regulation especially by c-Rel in the literature. Therefore, it is somewhat surprising to have it classified as a Rel-repressed gene here...what are the biological effects of the Myc-regulated genes?'

We are aware that Myc has been considered to be a Rel target gene in some studies. Indeed, we quoted these papers in our manuscript. The key insight here is that BOTH RelA and Rel can activate Myc transcription, but RelA is a better activator. Thus, Myc is expressed at higher levels in Rel-deficient B cells. In this sense, Myc is Rel-repressed.

- 11) 'A Comparison of present datasets with other publicly available datasets on B cell transcriptomes influenced by loss or gain of RelA or c-Rel is missing.'

We previously searched and have re-searched available datasets for time-dependent transcriptome analyses in B cells that lack RelA or Rel and did not find any. We will continue

to look for such datasets and will be happy if the reviewer could help identify the datasets he/she is referring to.

- 12) ‘How many cells are depicted in Figure S5a? Why are there such differences between RelA^{F/F} and WT samples? If these are due to batch effects, can they be corrected to allow an integrative analysis of the whole dataset?’

In the revised manuscript we now include cell numbers for each of the conditions for which single cell analysis was carried out (revised Figure S5a). Because our goal was to compare between control and RelA-deficient samples, we have not further probed the basis for difference between WT (C57BL/6) and *RelA^{f/f}* samples. The entire datasets had been uploaded to GEO for reviewer scrutiny at the time of first submission.

- 13) ‘As noted by the authors, the incomplete inactivation of RelA in CD19Cre RelA^{F/F} B cells affects their dataset. The authors could consider employing Mb1Cre instead to obtain much higher RelA knockout proportions.’

Reviewers 1 and 2 both raise the point about using a different Cre to re-do our analyses. But they do so from different perspectives. Reviewer 1’s concern about compensation during development would not be circumvented by the use of Mb1-cre as suggested by Reviewer 2. As described in responses to the Editor and Reviewer 1, we used Tat-Cre to delete *RelA* in mature B cells. Activation of such RelA-deficient cells closely paralleled responses in cells where *RelA* deletion was initiated by *Cd19-cre*.

Reviewer #3

This reviewer found our work to ‘address(es) the important problem of distinguishing selectivity in NF-kB signaling by focusing on the binding and transcriptional response of RelA and Rel in stimulated B cells.’

General comments

However, he/she ‘found the manuscript quite descriptive and less focused on particular questions.’ And was concerned that ‘it is not clear in its current state how this paper moves the field beyond what was already known. Many ChIP-seq papers have been published on NF-kB factors, and many studies of the temporal impact of NF-kB, but there is little discussion of these results.’

We had attempted to comprehensively cite the literature regarding earlier ChIP-Seq and transcriptome studies precisely to give due credit and minimize this kind of response. Obviously, we did not do a good job. Indeed, there are many earlier ChIP-Seq studies, however the majority of these were carried out in cell lines and fibroblasts in response to TNF α treatment. The closest comparisons with what we present here are in macrophages activated by LPS. Even

in these high-profile studies (which we quoted) we did not find use of multiple activation time points, use of both RelA and Rel antibodies, and combining ChIP-Seq with NF- κ B attenuation to identify direct NF- κ B targets. We believe that this tripartite approach is essential to identify target genes rigorously. As demonstrated by our studies, there are many RelA and Rel binding sites genome-wide that apparently do not confer changes in transcriptional activity. Additionally, we did not find temporal gene expression studies in B cells that coupled multiple genomic methodologies to address the fundamental question of which genes are direct targets and which genes are indirect targets (and therefore represent the NF- κ B-induced cascade).

Specific issues/concerns/comments

- 1) 'Page 6: Are the numbers of RelA and Rel peaks in keeping with what others have found? While I understand that the cell types may matter but Zhao et al. (PMID: 25159142), for example, found considerably more sites in resting lymphoblastoid cells (e.g., 20,067 RelA, 6,765 Rel peaks). There a number of other NF- κ B ChIP-seq datasets, so it would be helpful for the authors to comment on the numbers of peaks given the sensitivity to protocol and antibodies. Also, was the irreducible discovery rate (IDR) method (ENCODE consortium best practices) used for the replicate averaging?'

Our criteria for identifying ChIP-Seq peaks and incorporating biological replicates was spelt out in the Methods section and more briefly in the Results section. The question of number of peaks is tricky because it depends on what one calls a peak, which in turn depends on which peak calling software is used. As described in the manuscript we were especially conservative in this regard, calling RelA peaks as those that had a peak score greater than or equal to 10, and Rel peaks as those that had scores greater than or equal to 5. Furthermore, the numbers of peaks noted in Figure 1 referred ONLY to those peaks that were found in two independent ChIP-Seq replicates with the stringent peak score criterion. Did we miss some peaks? We probably did. However, those that we focus on in the manuscript represent rigorously identified RelA and Rel peaks in activated B cells. In most NF- κ B ChIP-Seq papers this kind information is not provided, making it difficult to directly compare peak numbers between studies. Despite using stringent criteria, we identified scores of target genes that had not been previously associated with NF- κ B. All ChIP-Seq datasets were provided to reviewers, thereby permitting future analyses with different parameters than ours. We point out that the Zhao et al. paper quoted by the reviewer used a transformed lymphoblastoid cell line for their studies and the details of peak calling, biological replicates etc are difficult to discern from the paper.

Based on the reviewer's suggestion we carried out IDR analysis on our ChIP-Seq samples and found all but one to adhered to ENCODE best practices definition. The one condition that did not pass IDR validation was RelA ChIP-Seq in unactivated B cells. This is likely because there are few robust RelA binding sites prior to BCR activation and it is not a time point that we emphasize in our analyses. We provide the IDR analysis for Reviewer evaluation but did not

incorporate it into Supplementary figures because we did not find that this was done for most published ChIP-Seq studies.

- 2) Page 6: Was motif finding performed on promoters only? It wasn't exactly clear from the methods.

Motif finding was carried for ALL RelA and Rel peaks. This has now been clarified in the text (page 6-7).

- 3) Page 6: The limitation to the top 3 motifs seems artificial. Towards a more encompassing and transparent analysis, I encourage the authors to list all significant motifs identified in their ChIP-seq peaks. If this becomes significantly more than 3 for each experiment, some form of a heatmap could be used to show them. I would also like to see (data can be in Supplement) that an independent motif analysis program (e.g., Centrimo from the MEME suite) arrives at similar conclusions for the motif analysis.

In the revised manuscript we provide a list of all significant motifs identified by HOMER (new Figure S1f and g).

Motif analysis was also carried out using TFmotifView (new Figure S1h and links provided therein).

- 4) Page 7: “We conclude that the first phase of NF-kB induced signaling leads to widespread recruitment of RelA genome-wide and more restricted recruitment of Rel to mostly overlapping sites.” Why is this ‘restricted’? In other words, is this just because it’s at lower concentration (i.e., it binds higher later). To help address this, it would be helpful if the authors could include some indication of the RelA and Rel protein (ideally nuclear) concentration along with the ChIP-seq peak data in Figure 1.

‘Restricted’ was used to indicate that Rel recruitment occurred at fewer sites compared to RelA recruitment at early time points. This could be due to lower Rel concentrations. To the best of our knowledge information about nuclear protein concentrations is not a part of any ChIP-Seq paper. We understand that this is more pertinent when two closely related factors are being simultaneously assayed but would defer this to future mechanistic studies on the various interesting observations made in the current manuscript.

- 5) Page 7: The authors note - “The IRF motif (alone or combined with PU.1) was not included in the top three RelA-specific binding regions” First, the syntax is confusing, it should probably not be ‘RelA-specific binding regions’, but ‘RelA-specific binding motifs’.

This has been clarified (page 7).

- 6) Page 7: “Promoter-associated RelA binding sites were were marked” – the word ‘were’ is repeated.

Thank you, the second 'were' has been deleted.

- 7) Page 7: “From time-dependent ATAC-Seq, we identified several thousand sites at which pre-existing chromatin accessibility was transiently increased ... We conclude that transiently bound RelA at these regions alters chromatin structure induced by constitutively expressed factors” While this seems reasonable, how do the authors determine when a difference in ATAC-seq peak is significant? Was a particular threshold for differential used? Further, how do the authors rule out the possibility that another TF is also binding to these peaks and leading to the observed changes?

Our manuscript contained a description of criteria used to identify significantly different ATAC peaks (fold change 1.5, FDR ≤ 0.05). We cannot rule out that another factor or factors bound to differentially accessible regions. Thus, the relationship between differential accessibility and RelA binding remains an association. We have changed the verbiage on page 8 (of the revised manuscript) to reflect the point that changes in ATAC sensitivity could also be caused by other inducible transcription factors.

- 8) Page 8: “The timing of gene expression changes mostly coincided with or followed changes in chromatin accessibility (Figure 3b).” Is this accessibility of the promoters? Please indicate in main text.

Accessibility changes were assessed for all ATAC peaks that were annotated to a gene by HOMER. Therefore, some inducible peaks shown in Figure 2c scored as being H3K4me3⁺ and could be attributed to promoters, whereas others that were H3K4me3⁻ likely represented inter- and intragenic regions.

- 9) Page 8: “We observed inducible RelA binding across 39% of genes in category ‘a’ (rapidly and transiently induced) and 13% of genes in categories ‘b’ and ‘c’ (Figure 3d).” What is meant by ‘binding across genes’? Is this the promoters of the genes? Across the gene body?

This point has been clarified in the text (page 9). Essentially, NF- κ B binding genes were identified based on annotation of RelA/Rel CHIP-Seq peaks to genes by HOMER which accounts for binding to promoters, intragenic and intergenic locations.

- 10) Page 9: “189 transiently induced RNAs were suppressed by the inhibitor (Figure 3e, cluster I3); 117 of these genes bound RelA at 1h, implicating them as direct NF- κ B targets (Table S1).” Again, does this mean the gene promoter was bound by RelA? Please re-word these as it is confusing what is meant by a gene binding to a TF.

As described in the response to #9 above, we refer not only to gene promoters that bind RelA, but to genes that have RelA binding as annotated by HOMER. In the revised manuscript we have altered the text to define how we annotated NF- κ B binding to genes (page 9).

- 11) Page 9: When is the BAY inhibitor given to the cells to identify the intermediate and late-phase activated genes? Is it just before that time point, or is it before induction? If it is before the IgM treatment, then it is not clear how the author rule out indirect effects of other late-phase TFs in the regulation of those target genes. For the 1h time-point it is similarly a potential issue, but perhaps less so.

As noted in the manuscript, BAY inhibitor was administered prior to activation (anti-IgM treatment). The experimental design was based on the idea that BAY inhibits phase 1 NF- κ B (both ReA and Rel) and thereby all consequences of NF- κ B activation. Some of these are direct effects on NF- κ B target genes and the rest are indirect effects on NF- κ B-induced transcription factors. The reviewer correctly points out that BAY may inhibit other currently unknown transcription factors. To circumvent precisely this concern, we focused on genes that were affected by BAY and bound either RelA (early) or Rel (late). The effects of BAY even on NF- κ B binding genes may be mediated by other transcription factors. To address this, we distinguished between direct and indirect effects of NF- κ B using ChIP-Seq data and transcriptional studies in RelA-cKO and Rel-deficient B cells (which were not treated with BAY).

- 12) Page 10: “Absence of Rel also increased inducible activation of many genes at 1h (Figure 3g, cluster R2), suggesting widespread repressive role for this family” This conclusion is complicated as there could be a number of compensatory mechanisms that result in the cells that have Rel knocked out. To confirm this conclusion it would be important to show that over-expression of cRel (ideally in an inducible manner) suppressed these genes.
- 13) Page 11: “To the best of our knowledge, this is the first demonstration of the differential activation potential of RelA or Rel on endogenous genes.” As above, this interpretation is complicated by the possible compensatory mechanisms in the knockout cells. Over expressing each TF in an inducible manner – and showing that it leads to opposite effects on these genes – would seem to more directly address this point of activation potential. This is an intriguing idea, and one that appears to occur in a gene/loci-specific manner, so a more direct experiment would be very helpful to support it.

These last two points refer to RelA/Rel antagonism; the reviewer indicates the need to establish sufficiency of the effects attributed to each factor by over-expressing each in an inducible manner. It is true that by using factor deficiency to make the case, we have only shown the necessity of each factor and not sufficiency. Moreover, it is not our intention to claim that one of these proteins is a repressor, rather our idea is that both can activate a subset of genes at early time points, but RelA is a stronger activator compared to Rel (page 12, middle paragraph). Additional mechanistic studies are, in our opinion, beyond the range of this manuscript.

3.5 decades of NF- κ B research have defined complex signaling pathways and equally complex physiological phenotypes for genetic knockouts of Rel family members. The one area

that we believe is under-developed is that of critically defining genes whose expression is affected by this transcription factor. That is, transcriptional consequences of NF- κ B activation. Thus, our primary objective in carrying out the studies described in this manuscript was to identify RelA- and Rel target genes in BCR-activated B cells. Our approach identified all the 'usual suspects' and many novel genes not previously attributed to being under NF- κ B control. While achieving this goal several interesting biological insights were uncovered, such as defining possible targets of Rel in GC function, defining candidate NF- κ B targets involved in lymphomagenesis, revealing heterogeneity of the NF- κ B response and providing evidence for functional antagonism between RelA and Rel at early activation time points. Additionally, based on a reviewer suggestion, we also demonstrated distinct physiological functions of early and late NF- κ B in BCR-activated B cells. Our study design can be readily adapted to the many cell types and signals where NF- κ B function is implicated to illuminate NF- κ B initiated gene expression cascades necessary to understand its functional contributions.

We believe we have made a good faith effort to address all editor and reviewer comments and suggestions, and hope you will find the revised manuscript suitable for publication in *Nature Immunology*.

Sincerely,

Ranjan Sen, Ph.D.
Chief, Laboratory of Molecular Biology and
Immunology

Figure a - Irreproducibility Discovery Rate (IDR) analysis for ChIP-Seq replicates.

RelA0h
Reproducibility QC and peak detection statistics

	overlap	idr
Nt	57816	3051
N1	49083	6347
N2	31103	1132
Np	65232	6273
N optimal	65232	6273
N conservative	57816	3051
Optimal Set	pooled-pr1_vs_pooled-pr2	pooled-pr1_vs_pooled-pr2
Conservative Set	rep1_vs_rep2	rep1_vs_rep2
Rescue Ratio	1.12826899128269	2.056047197640118
Self Consistency Ratio	1.5780792849564351	5.6068904593639575
Reproducibility Test	pass	fail

RelA1h
Reproducibility QC and peak detection statistics

	overlap	idr
Nt	75841	6592
N1	68905	9068
N2	60125	3305
Np	79698	9208
N optimal	79698	9208
N conservative	75841	6592
Optimal Set	pooled-pr1_vs_pooled-pr2	pooled-pr1_vs_pooled-pr2
Conservative Set	rep1_vs_rep2	rep1_vs_rep2
Rescue Ratio	1.0508563969356943	1.3968446601941749
Self Consistency Ratio	1.1460291060291061	2.7437216338880486
Reproducibility Test	pass	borderline

RelA4h
Reproducibility QC and peak detection statistics

	overlap	idr
Nt	50599	3783
N1	29572	3019
N2	43278	1586
Np	53051	4190
N optimal	53051	4190
N conservative	50599	3783
Optimal Set	pooled-pr1_vs_pooled-pr2	pooled-pr1_vs_pooled-pr2
Conservative Set	rep1_vs_rep2	rep1_vs_rep2
Rescue Ratio	1.0484594557204687	1.1075865715040973
Self Consistency Ratio	1.4634789665900176	1.903530895334174
Reproducibility Test	pass	pass

Rel0h
Reproducibility QC and peak detection statistics

	overlap	idr
Nt	33625	356
N1	4263	81
N2	17783	117
Np	31386	425
N optimal	33625	425
N conservative	33625	356
Optimal Set	rep1_vs_rep2	pooled-pr1_vs_pooled-pr2
Conservative Set	rep1_vs_rep2	rep1_vs_rep2
Rescue Ratio	1.0713375390301407	1.1938202247191012
Self Consistency Ratio	4.171475486746423	1.4444444444444444
Reproducibility Test	borderline	pass

Rel1h
Reproducibility QC and peak detection statistics

	overlap	idr
Nt	35761	1417
N1	11839	523
N2	4118	310
Np	40861	1400
N optimal	40861	1417
N conservative	35761	1417
Optimal Set	pooled-pr1_vs_pooled-pr2	rep1_vs_rep2
Conservative Set	rep1_vs_rep2	rep1_vs_rep2
Rescue Ratio	1.1426134615922374	1.0121428571428572
Self Consistency Ratio	2.874939290917921	1.6870967741935483
Reproducibility Test	borderline	pass

Rel18h
Reproducibility QC and peak detection statistics

	overlap	idr
Nt	55164	3179
N1	45168	1387
N2	43075	1744
Np	54830	3034
N optimal	55164	3179
N conservative	55164	3179
Optimal Set	rep1_vs_rep2	rep1_vs_rep2
Conservative Set	rep1_vs_rep2	rep1_vs_rep2
Rescue Ratio	1.0060915557176728	1.0477916941331575
Self Consistency Ratio	1.0485896691816599	1.2573900504686373
Reproducibility Test	pass	pass

Decision Letter, first revision:

6th Apr 2023

Dear Ranjan,

We re-reviewed your manuscript entitled "NF-κB subunits direct kinetically distinct transcriptional cascades in antigen receptor-activated B cells", reference number NI-A33546A. As noted in my previous message, although the referees expressed that there was nothing technically wrong with the study, in their opinion it did not represent a sufficient conceptual advance for publication in Nature Immunology.

Please note, I have discussed your manuscript with my colleague Meriem Attaf at Nature Communications. She has expressed interest in your work and would be willing to accept your manuscript for publication in Nature Communications based on the reviewer reports from Nature Immunology.

Should you decide to transfer this manuscript to Nature Communications, please include a new cover

letter addressed to Meriem Attaf. If any major changes are made to the manuscript prior to transfer, you may wish to highlight this in the cover letter. In addition, please also send an email to meriem.ataf@nature.com to communicate your Nature Communications manuscript reference number to her.

If you would like further information in order to make the decision to transfer, please do not hesitate to contact Meriem directly. Otherwise, please use the link below to transfer the manuscript to Nature Communications.

We regret this outcome, but appreciate your continued interest in Nature Immunology. I am sure that at some future date our interactions will end on a more positive note.

Kind regards,

Laurie

Laurie A. Dempsey, Ph.D.
Senior Editor
Nature Immunology
l.dempsey@us.nature.com
ORCID: 0000-0002-3304-796X

Reviewers' comments:

Reviewer #1 (Remarks to the Author):

In the revised version, authors add several experiments, according to reviewers' suggestions. I think that this is nice, but I think that the revised version still does not provide novel aspects in the NF- κ B area.

Reviewer #2 (Remarks to the Author):

In their revised manuscript Zhao and colleagues further improved mostly technical aspects of their work. In my opinion, this revised version does not address my main criticism/concerns regarding the lack of novel biological or mechanistic insights.

The most significant advance of this revision is in my view the differentiation between marginal zone and follicular B cells. Here the authors describe attenuated early NF- κ B in marginal zone B cells, which appears to be at odds with the pre-activated status of these cells. However, the consequences and potential meaning of this finding is not discussed. The use of TatCre to acutely ablate RelA in mature B cells addressed concerns regarding developmental compensation in the RelA-deficient B lineage in vivo.

My main original criticism was that the authors do not leverage their findings to biologically relevant processes in B cell biology (original points 1, 2, 8). They now provide KEGG pathway analysis yielding very general terms such as "TNF signaling", "IL-17 signaling", apoptosis and various cancer-related

pathways cell cycle regulation, metabolism and neurodegenerative diseases". I fail to see how this significantly advances our understanding of B cell activation or transformation beyond the original version of the manuscript.

All suggestions (original points 3, 4) to leverage their findings into mechanistic insight were delegated to future studies. I fully agree with the authors that a manuscript can be focused on one particular aspect, which here is targets of early and late NF-kappaB activation in B cells. Also, one important aspect of the present work is to provide a resource for the community. However, the authors should provide clear exemplary evidence that (and how) their resource can be exploited to gain novel insights into the biology of B cell responses by providing such insights.

Some of the other additional experiments seem rather limited to me. For example, it would have been interesting and quite straightforward to assess the nuclear translocation of c-Rel and RelA simultaneously (original point 6) over the time-course of the manuscript at the single B cell level (instead of RelA following 1h of aIgM). In addition, the target validation was conducted with the IKK2 inhibitor, but the investigation of RelA- and c-Rel-deficient B cells would have been more relevant to the study.

Examples for publicly available datasets on B cell transcriptomes influenced by loss or gain.

Cited by the authors:

Heise, N., Silva, N.S.D., Silva, K., Carette, A., Simonetti, G., Pasparakis, M., and Klein, U. (2014). Germinal center B cell maintenance and differentiation are controlled by distinct NF- κ B transcription factor subunits. *J Exp Med* 211, 2103–2118 (ex vivo GCB cells c-Rel knockout, in vitro, CD40 + aIgM activated B cells, RelA and c-Rel knockout)

Kober-Hasslacher, M., Oh-Strauß, H., Kumar, D., Soberon, V., Diehl, C., Lech, M., Engleitner, T., Katab, E., Fernández-Sáiz, V., Piontek, G., Li, H., Menze, B., Ziegenhain, C., Enard, W., Rad, R., Böttcher, J.P., Anders, H.-J., Rudelius, M., and Schmidt-Supprian, M. (2020). c-Rel gain in B cells drives germinal center reactions and autoantibody production. *Journal of Clinical Investigation* 130, 3270–3286 (ex vivo GCB, naive B and plasma cells, c-Rel overexpression).

Others:

Mouse Myc-driven lymphoma ex vivo analyzed by microarray: Hunter, J.E., Butterworth, J.A., Zhao, B., Sellier, H., Campbell, K.J., Thomas, H.D., Bacon, C.M., Cockell, S.J., Gewurz, B.E., and Perkins, N.D. (2015). The NF- κ B subunit c-Rel regulates Bach2 tumour suppressor expression in B-cell lymphoma. *Oncogene*.

Zhao, B., Barrera, L.A., Ersing, I., Willox, B., Schmidt, S.C.S., Greenfeld, H., Zhou, H., Mollo, S.B., Shi, T.T., Takasaki, K., Jiang, S., Cahir-McFarland, E., Kellis, M., Bulyk, M.L., Kieff, E., and Gewurz, B.E. (2014). The NF- κ B Genomic Landscape in Lymphoblastoid B Cells. *Cell Reports* 8, 1595–1606.

Human Hodgkin Lymphoma: Oliveira, K.A.P. de, Kaergel, E., Heinig, M., Fontaine, J.-F., Patone, G., Muro, E.M., Mathas, S., Hummel, M., Andrade-Navarro, M.A., Hübner, N., and Scheidereit, C. (2016). A roadmap of constitutive NF- κ B activity in Hodgkin lymphoma: Dominant roles of p50 and p52 revealed by genome-wide analyses. *Genome Medicine* 8, 28.

Human DLBCL, c-Rel-associated gene signature: Faumont, N., Taoui, O., Collares, D., Jais, J.-P., Leroy, K., Prévaud, L., Jardin, F., Molina, T.J., Copie-Bergman, C., Petit, B., Gourin, M.-P., Bordessoule, D., Troutaud, D., Baud, V., and Feuillard, J. (2021). c-Rel Is the Pivotal NF- κ B Subunit in Germinal Center Diffuse Large B-Cell Lymphoma: A LYSA Study. *Frontiers Oncol* 11, 638897.

Reviewer #3 (Remarks to the Author):

The authors have addressed my previous concerns.

Author Rebuttal, first revision:

Hi Laurie:

Before I resort to down-grading our manuscript to *Nat. Comm.* I wanted to run the following thoughts by you.

This manuscript was originally being considered as a Resource publication. While I believe that our work has novel aspects, I suspect that conceptual novelty carries lower weight in a Resource article. So, I remain a tad surprised that the reviewers after being fairly complementary in the first review and asking for a whole bunch of new data, which we provided, came back with a lack-of-novelty verdict. Do they really think that the data (time courses of RNA-Seq in WT and Rel and RelA-deficient B cells, time courses of ChIP-Seq with different Rel family members, time courses of ATAC-Seq, time courses of single cell RNA-Seq in Rel- and RelA-deficient B cells, and finally time courses with purified MZ and Fo B cells) is not a unique resource for the community? I know of no other published data that approaches this level. Some time ago NI published a study with many genomic assays in activated B cells, but it was at a late activation time point. I think it is quite difficult to mechanistically unravel what happens at 72h without knowing how cells respond at early time points, emphasizing the need for the kinds of observations presented in our manuscript.

None of the above thoughts diminish my deep appreciation for your having reached out to the folks at *Nat. Comm.* Let me know what you think and I will proceed accordingly.

Best,

ranjan

Decision Letter, second revision:

Our ref: NI-A33546B-Z

4th May 2023

Dear Ranjan,

Apologies for the delay in getting back to you and your request to reconsider your manuscript entitled "NF- κ B subunits direct kinetically distinct transcriptional cascades in antigen receptor-activated B cells" (NI-A33546B-Z).

I discussed the study and the previous reviews of the manuscript with the others on the editorial team. Given that the referees do not have technical comments on the experiments or the datasets obtained, nor on the strength of conclusions based on your new data, but only on their opinion on 'how much of an advance' does the study provide with respect to NF- κ B, we decided to overturn our decision. Therefore we'll be happy in principle to publish it in Nature Immunology, pending minor revisions to comply with our editorial and formatting guidelines.

We will now perform detailed checks on your paper and will send you a checklist detailing our editorial and formatting requirements in about a week. Please do not upload the final materials and make any revisions until you receive this additional information from us.

If you had not uploaded a Word file for the current version of the manuscript, we will need one before beginning the editing process; please email that to immunology@us.nature.com at your earliest convenience.

Thank you again for your interest in Nature Immunology Please do not hesitate to contact me if you have any questions.

Kind regards,

Laurie

Laurie A. Dempsey, Ph.D.
Senior Editor
Nature Immunology
l.dempsey@us.nature.com
ORCID: 0000-0002-3304-796X

Final Decision Letter:

Dear Ranjan,

I am delighted to accept your manuscript entitled "NF- κ B subunits direct kinetically distinct transcriptional cascades in antigen receptor-activated B cells" for publication in an upcoming issue of Nature Immunology.

Over the next few weeks, your paper will be copyedited to ensure that it conforms to Nature Immunology style. Once your paper is typeset, you will receive an email with a link to choose the appropriate publishing options for your paper and our Author Services team will be in touch regarding any additional information that may be required.

Please note that *Nature Immunology* is a Transformative Journal (TJ). Authors may publish their research with us through the traditional subscription access route or make their paper immediately open access through payment of an article-processing charge (APC). Authors will not be required to make a final decision about access to their article until it has been accepted. [Find out more about Transformative Journals](https://www.springernature.com/gp/open-research/transformative-journals).

Authors may need to take specific actions to achieve [compliance with funder and institutional open access mandates](https://www.springernature.com/gp/open-research/funding/policy-compliance-faqs). If your research is supported by a funder that requires immediate open access (e.g. according to [Plan S principles](https://www.springernature.com/gp/open-research/plan-s-compliance)) then you should select the gold OA route, and we will direct you to the compliant route where possible. For authors selecting the subscription publication route, the journal's standard licensing terms will need to be accepted, including [self-archiving policies](https://www.springernature.com/gp/open-research/policies/journal-policies). Those licensing terms will supersede any other terms that the author or any third party may assert apply to any version of the manuscript.

Your paper will be published online soon after we receive your corrections and will appear in print in the next available issue. Content is published online weekly on Mondays and Thursdays, and the embargo is set at 16:00 London time (GMT)/11:00 am US Eastern time (EST) on the day of publication. Now is the time to inform your Public Relations or Press Office about your paper, as they might be interested in promoting its publication. This will allow them time to prepare an accurate and satisfactory press release. Include your manuscript tracking number (NI-A33546C) and the name of

the journal, which they will need when they contact our office.

About one week before your paper is published online, we shall be distributing a press release to news organizations worldwide, which may very well include details of your work. We are happy for your institution or funding agency to prepare its own press release, but it must mention the embargo date and Nature Immunology. Our Press Office will contact you closer to the time of publication, but if you or your Press Office have any enquiries in the meantime, please contact press@nature.com.

Also, if you have any spectacular or outstanding figures or graphics associated with your manuscript - though not necessarily included with your submission - we'd be delighted to consider them as candidates for our cover. Simply send an electronic version (accompanied by a hard copy) to us with a possible cover caption enclosed.

If you have not already done so, we strongly recommend that you upload the step-by-step protocols used in this manuscript to the Protocol Exchange. Protocol Exchange is an open online resource that allows researchers to share their detailed experimental know-how. All uploaded protocols are made freely available, assigned DOIs for ease of citation and fully searchable through nature.com. Protocols can be linked to any publications in which they are used and will be linked to from your article. You can also establish a dedicated page to collect all your lab Protocols. By uploading your Protocols to Protocol Exchange, you are enabling researchers to more readily reproduce or adapt the methodology you use, as well as increasing the visibility of your protocols and papers. Upload your Protocols at www.nature.com/protocolexchange/. Further information can be found at www.nature.com/protocolexchange/about .

Please note that we encourage the authors to self-archive their manuscript (the accepted version before copy editing) in their institutional repository, and in their funders' archives, six months after publication. Nature Portfolio recognizes the efforts of funding bodies to increase access of the research they fund, and strongly encourages authors to participate in such efforts. For information about our editorial policy, including license agreement and author copyright, please visit www.nature.com/ni/about/ed_policies/index.html

Kind regards,

Laurie

Laurie A. Dempsey, Ph.D.
Senior Editor
Nature Immunology
l.dempsey@us.nature.com
ORCID: 0000-0002-3304-796X